# Selective photocatalytic $CO_2$ reduction in aerobic environment by microporous Pd-porphyrin-based polymers coated hollow $TiO_2$

Yajuan Ma[1], Xiaoxuan Yi[1], Shaolei Wang[2], Tao Li[1], Bien Tan [1], Chuncheng Chen[3], Tetsuro Majima[1], Eric R. Waclawik [4], Huaiyong Zhu [4] & Jingyu Wang [1]✉

Direct photocatalytic $CO_2$ reduction from primary sources, such as flue gas and air, into fuels, is highly desired, but the thermodynamically favored $O_2$ reduction almost completely impedes this process. Herein, we report on the efficacy of a composite photocatalyst prepared by hyper-crosslinking porphyrin-based polymers on hollow $TiO_2$ surface and subsequent coordinating with Pd(II). Such composite exhibits high resistance against $O_2$ inhibition, leading to 12% conversion yield of $CO_2$ from air after 2-h UV-visible light irradiation. In contrast, the $CO_2$ reduction over Pd/$TiO_2$ without the polymer is severely inhibited by the presence of $O_2$ ($\geq 0.2$ %). This study presents a feasible strategy, building Pd(II) sites into $CO_2$-adsorptive polymers on hollow $TiO_2$ surface, for realizing $CO_2$ reduction with $H_2O$ in an aerobic environment by the high $CO_2$/$O_2$ adsorption selectivity of polymers and efficient charge separation for $CO_2$ reduction and $H_2O$ oxidation on Pd(II) sites and hollow $TiO_2$, respectively.

[1] Key Laboratory of Material Chemistry for Energy Conversion and Storage (Ministry of Education), School of Chemistry and Chemical Engineering, Huazhong University of Science and Technology, Wuhan 430074, China. [2] Key Laboratory of Polyoxometalate Science of Education Institution, Faculty of Chemistry, Northeast Normal University, Changchun 130024, China. [3] Key Laboratory of Photochemistry, CAS Research/Education Center for Excellence in Molecular Sciences, Institute of Chemistry, Chinese Academy of Sciences, Beijing 100190, China. [4] School of Chemistry, Physics and Mechanical Engineering, Queensland University of Technology, Brisbane QLD 4001, Australia. ✉email: wangjingyu@hust.edu.cn

Photocatalytic $CO_2$ reduction into useful fuels is a promising approach to tackle the challenges of carbon emission and global warming by directly utilizing sustainable solar energy[1–3]. Despite extensive efforts and many attempts at harnessing various semiconductor photocatalysts for $CO_2$ reduction, most of the photocatalytic reactions occur only at high $CO_2$ concentration and sometimes $CO_2$-philic organic solvents are required to make them operate efficiently, due to low $CO_2$ uptake of the photocatalysts[1–6]. Physisorptive microporous solids such as microporous organic polymers and metal-organic frameworks have recently emerged as promising candidates to replace aqueous amines for $CO_2$ capture and storage[7–11]. Several reports demonstrated the integration of metals into $CO_2$-adsorptive materials could convert diluted $CO_2$ due to the high $CO_2$ uptake and high reduction activity of metals during the photocatalytic reactions[12–14]. However, metal sites suffer from poor $H_2O$ oxidation activity and highly active for $H_2$ evolution from $H_2O$, so that these photocatalysts require addition of Ru-containing photosensitizer together with organic sacrificial reagent and solvent[12–14], which present unsustainable and negative environmental impact issues. More importantly, anaerobic environment is essential to avoid the competitive reaction of oxygen reduction because it is thermodynamically favored compared to $CO_2$ reduction[15–20].

An ideal catalyst is capable of taking gaseous feedstocks[21]. In practice, the $CO_2$ concentration in air is as low as 300~400 ppm, and flue gas after fossil fuel combustion typically consists of about 72–77 vol% $N_2$, 12–14 vol% $CO_2$, 8–10 vol% $H_2O$, 3–5 vol% $O_2$, and other minor components[21–24]. In air and flue gas, the $CO_2$ adsorption and activation on the surface of photocatalysts are low, due to the competitive $O_2$ adsorption and reduction, as well as the low $CO_2$ concentration[15–20]. Catalytic $CO_2$ reduction is strongly influenced by the presence of 5 ppm of $O_2$ and completely inhibited in 5 vol% $O_2$, because $O_2$ reduction is thermodynamically favored compared to $CO_2$ reduction[20]. Therefore, to control $CO_2$ emission from exhaust gas and reduce $CO_2$ concentration in air, developing efficient photocatalysts with selective $CO_2$ adsorption and conversion in an aerobic environment remains a challenge.

To address the challenge, we envisioned that significantly increasing $CO_2$ concentration around the catalytic active sites for $CO_2$ reduction via preferential adsorption of $CO_2$ over $O_2$ could lessen the inhibitive impact of $O_2$ and promoting $H_2O$ oxidation could increase $CO_2$ conversion. High $CO_2$ adsorption capability and selectivity of microporous polymers with heterocyclic skeleton and large π-conjugated structure can bring opportunities for directly using low concentration of $CO_2$ without separation from aerobic mixtures if the catalytic active sites for $CO_2$ reduction are built in the polymer. For the photocatalytic $CO_2$ reduction with $H_2O$ in such an aerobic environment, another essential requirement is to assemble the photocatalytic sites for $CO_2$ reduction and $H_2O$ oxidation for efficient separation of photogenerated electrons and holes, respectively. Meanwhile, $H_2O$ provides protons for reacting with the intermediates from $CO_2$ reduction, increasing $CH_4$ production. Electron transfer at the heterointerface between two components is required for the occurrence of $CO_2$ reduction and $H_2O$ oxidation at different active sites in a composite structure.

In this work, a proof-of-concept study was conducted to verify this hypothesis. We prepared a porous composite photocatalyst by in situ hyper-crosslinking porphyrin-based polymers (HPP) on a hollow $TiO_2$ surface, followed by loading Pd(II) via coordination with HPP to form the $CO_2$ reduction sites (Pd-HPP-$TiO_2$). Hollow $TiO_2$ was used to increase the heterointerface between $TiO_2$ and Pd-HPP. The choice of Pd allows us to confirm the influence of $CO_2$ adsorption and charge separation on the reduction. The heteroatom-rich microporous structure can not only improve the capability and selectivity of $CO_2$ adsorption in an aerobic environment but also stabilize Pd(II) sites, while anatase $TiO_2$ surface is highly efficient for $H_2O$ oxidation with holes that generated in the valence band from the bandgap excitation. Pd-HPP-$TiO_2$ achieves the efficient conversion of $CO_2$ in an aerobic environment, i.e., 12 % of $CO_2$ in air is converted after 2-h UV-visible light irradiation with a $CH_4$ production of 24.3 μmol g$^{-1}$, which is 4.5 times higher than that over Pd/$TiO_2$. Based on the catalytic activity, we identify the active sites for photocatalytic $CO_2$ reduction and discuss the overall reaction mechanism.

## Results and discussion

**Preparation of porous Pd-HPP-TiO₂.** The synthetic processes of porous Pd-HPP-$TiO_2$ are depicted in Fig. 1. HPP were knitted

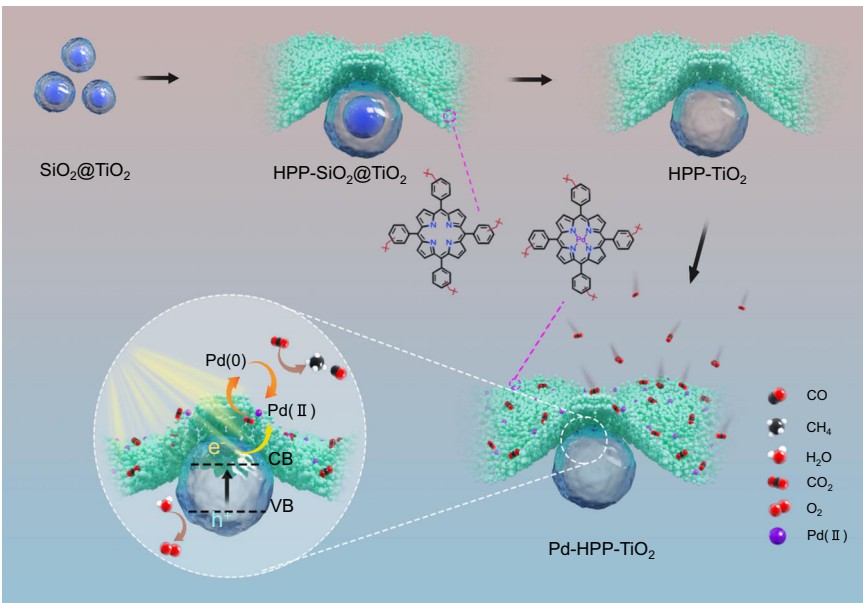

**Fig. 1 Schematic illustrations.** Synthesis of porous Pd-HPP-$TiO_2$ and the possible mechanism of photocatalytic $CO_2$ reduction. The chemical structures of HPP and Pd-HPP units are provided.

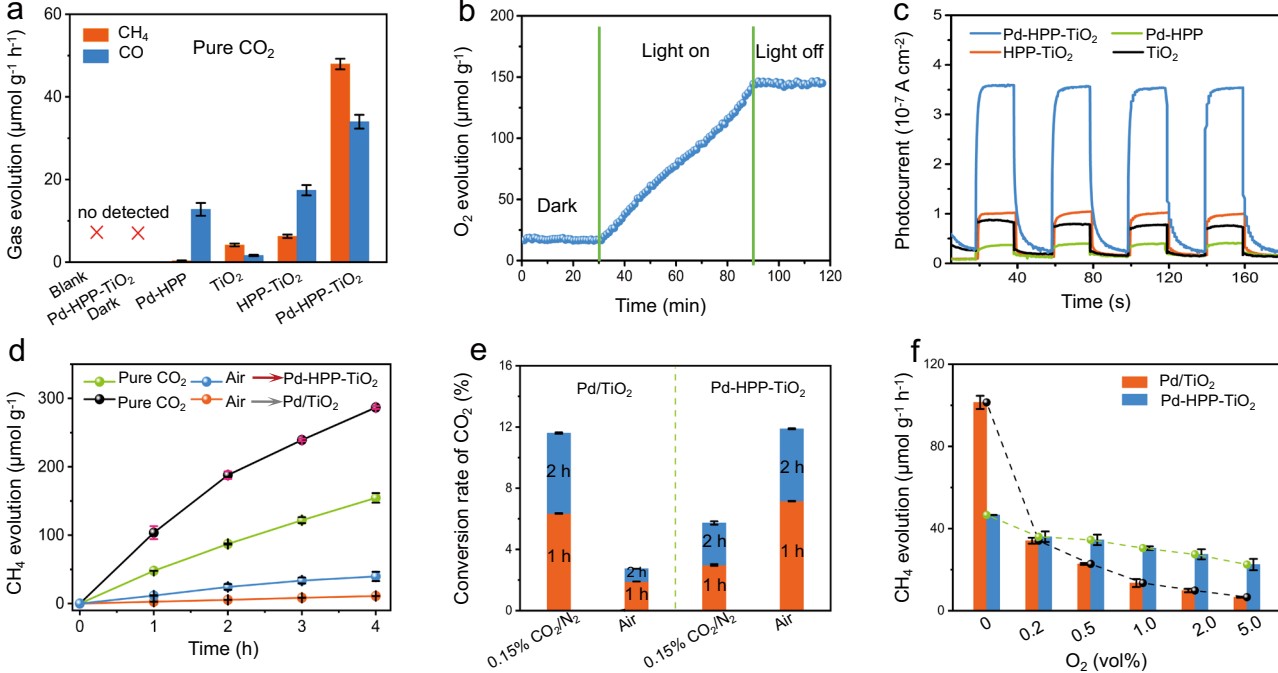

**Fig. 2 Evaluation of photocatalytic CO₂ reduction. a** The evolution rates of CH₄ and CO in pure CO₂. **b** On-line monitoring of O₂ evolution during the photocatalytic reaction over Pd-HPP-TiO₂. **c** Photocurrent response during light on-off cycling. **d** Comparison of the CH₄ evolution rates in pure CO₂ and air. **e** The conversion yield of CO₂ by measuring the CO₂ concentration. **f** Effect of O₂ concentration (vol%) in CO₂/O₂ gas mixture on the CH₄ evolution rate. The results in (**a**, **d**, **e**, **f**) are the average values of three parallel experiments. The error bar represents the standard deviation of the measurements.

together from 5,10,15,20-tetraphenylporphyrin (TPP) building blocks on the surface of core-shell $SiO_2@TiO_2$ with the diameter of 100–150 nm used as solid templates. The $SiO_2$ cores were then etched by NaOH solution to produce hollow $TiO_2$ with the thickness of about 10 nm coated by layers of HPP with the thickness of about 5–7 nm (HPP-$TiO_2$), finally Pd(II) coordinates with the core of porphyrin unit, leading to the formation of porous Pd-HPP-$TiO_2$ (details are provided in the Methods). The photocatalytic activity was evaluated in a gas-solid reaction without the addition of photosensitizer or organic sacrificial reagent under UV-visible light irradiation. $CH_4$ and CO were detected as the main products, in accordance with the results from many gas-solid reactions[25,26]. To clarify the effect of HPP, Pd/$TiO_2$ was synthesized as a control by photo-deposition of Pd nanoparticles on the surface of hollow $TiO_2$.

**Photocatalytic CO₂ reduction**. Figure 2a shows the comparison of $CH_4$ and CO evolution rates over a series of photocatalysts in pure $CO_2$. Hollow $TiO_2$ presented evolution rates of 4.2 and 1.6 μmol g⁻¹ h⁻¹ for $CH_4$ and CO, respectively. HPP-$TiO_2$ caused a moderate increase of CO evolution rate, mainly arising from the introduction of an abundance of micropores on HPP, favoring the $CO_2$ uptake. When building Pd(II) sites into HPP-$TiO_2$, the $CO_2$ reduction efficiency was further enhanced, reaching high evolution rates of 48.0 and 34.0 μmol g⁻¹ h⁻¹ (average value within 4 h) for $CH_4$ and CO, respectively. The comparison to the reported results under similar reaction conditions suggests the excellent photocatalytic activity of porous Pd-HPP-$TiO_2$ composite (Supplementary Table 1). The high selectivity as 59% for $CH_4$ production over Pd-HPP-$TiO_2$ can be attributed to the Pd(II) sites with sufficient energy overcoming the Schottky barrier with $TiO_2$ and improving the charge separation efficiency[27]. In a long-term test, Pd-HPP-$TiO_2$ showed continuous $CH_4$ and CO production up to 20 h under UV-visible light irradiation (Supplementary Fig. 1). Although there is

somewhat loss in catalytic activity, the superior performance of porous Pd-HPP-$TiO_2$ to Pd/$TiO_2$ during long-term photocatalytic reaction suggests that the introduction of microporous HPP greatly contributes to stabilizing Pd(II) sites. No detectable $H_2$ during the photocatalytic reaction suggests the higher $CO_2$ reduction selectivity than $H_2O$ reduction. It is noted that Pd-HPP exhibits the ability to catalyze the conversion of $CO_2$ to CO, which is consistent with the reports on various metal complexes[12–14,28]. Only $CH_4$ was produced over the photocatalysts containing $TiO_2$. The surface of anatase $TiO_2$ efficiently adsorbs $H_2O$ to facilitate $H_2O$ oxidation and provide protons for the $CO_2$ reduction to yield $CH_4$[29–31]. The efficient consumption of light-generated holes on the $TiO_2$ surface can accelerate the overall reaction.

The photocatalytic $CO_2$ reduction was confirmed by a series of control experiments, (1) dark reaction, (2) without photocatalyst, (3) in $N_2$, and (4) isotopic label using $^{13}CO_2$. No detectable product in the dark or the absence of photocatalysts indicates that the $CO_2$ reduction proceeded as a light driven catalytic process (Fig. 2a). Upon replacing $CO_2$ with $N_2$, trace amounts of $CH_4$ and CO were detected after the photocatalytic reaction (Supplementary Fig. 2), presumably due to the slight decomposition of HPP and the presence of pre-adsorbed $CO_2$ on the photocatalyst or reactor surface. Isotopically labeled $^{13}CO_2$ ($^{13}C$ enrichment of ≥ 97%) was used as the reactant to study the origin of products. According to the ion fragment analysis, the peaks at 2.5 min and 7.2 min in the gas chromatography could be assigned to $CH_4$ and CO, respectively. As compared to the signals of products under $^{12}CO_2$, the appearance of ion fragment peaks at m/z = 17 and 29 reveals that the produced $^{13}CH_4$ and $^{13}CO$ originated from $^{13}CO_2$ reduction over Pd-HPP-$TiO_2$ (Supplementary Fig. 3). The overall reaction involves $CO_2$ reduction and $H_2O$ oxidation cycles to produce $CH_4$, CO, and $O_2$, respectively, but their concentration changes in the whole cycle have seldom been measured in the literature[32]. The in-situ monitoring of $O_2$ evolution during the

photocatalytic reaction was performed to further verify the $CO_2$ reduction by $H_2O$ (Fig. 2b). When the experiment was conducted in the dark, low $O_2$ concentration in the reaction remained unchanged and came from the residual air, which was not completely removed by the degassing procedure. By way of contrast, during the photocatalytic reaction, the concentration of $O_2$ increased linearly with an evolution rate of 127 μmol $g^{-1}$ $h^{-1}$. Thus, the electrons ($e^-$) being provided from the 4-$e^-$ oxidation are comparable to the total electrons for $CH_4$ and CO production via 8-$e^-$ and 2-$e^-$ reduction processes. When UV-visible light irradiation was turned off, the $O_2$ concentration remained constant. This result illustrates that $CO_2$ is reduced by $H_2O$ during photocatalytic reaction over Pd-HPP-$TiO_2$ and that the backward reaction does not take place.

As for the photocatalytic $CO_2$ reduction, the reaction involved several steps, light absorption to generate electrons and holes in $TiO_2$, electron transfer from the conduction band of $TiO_2$ to Pd-HPP, electron trapping at the catalytic Pd(II) sites in Pd-HPP, reduction of the adsorbed $CO_2$ on Pd(II) sites[33]. The charge separation and charge transfer efficiency were investigated by electrochemical, photochemical, and photoelectrochemical measurements. The hollow $TiO_2$ displays a large semicircle arc at the high frequency of electrochemical impedance spectrum (EIS), indicating less electronic conductivity and larger electron transfer resistance ($R_{ct}$) (Supplementary Fig. 4). The value of $R_{ct}$ in the EIS of HPP-$TiO_2$ is smaller than those of both $TiO_2$ and Pd-HPP, indicating that the interface between the polymer and $TiO_2$ facilitates the electron transfer. The smallest value of $R_{ct}$ in the EIS of Pd-HPP-$TiO_2$ illustrates the efficient interfacial electron transfer by the surface binding HPP coordinating with Pd on hollow $TiO_2$. When coating with Pd-HPP, the photoluminescence of $TiO_2$ was almost quenched, indicating the efficient suppression of photogenerated charge recombination through radiative pathways (Supplementary Fig. 5). The photogenerated electrons are expected to transfer from the photoexcited $TiO_2$ to Pd(II) sites in HPP, leading to effective separation of electrons from the holes left in $TiO_2$. In addition, the introduction of Pd enhanced the interaction with gas molecules such as $O_2$ and $CO_2$ from air, which also causes the quenching of photoluminescence on $TiO_2$ surface[34–36]. Similar results for Pd/$TiO_2$ to Pd-HPP-$TiO_2$ in Supplementary Figs. 4 and 5 reveal that Pd in Pd-HPP contributes to the efficient charge separation. The amperometric signals provide further information on the relative efficiency of the electron transfer in the materials under UV-visible light irradiation. The electronic conductivity of Pd-HPP appears to be higher than $TiO_2$, and Pd-HPP exhibits the lowest photocurrent among the samples (Fig. 2c). Generally, the photoinduced charge separation in organic polymers does not occur dominantly as compared with exciton migration, leading to the lower capability as redox photocatalysts[37]. Although Pd-HPP possesses strong absorption in the visible region (Supplementary Fig. 6), both Pd-HPP and Pd-HPP-$TiO_2$ exhibit low photocatalytic activity for $CO_2$ reduction under visible light irradiation, suggesting the low efficiency of charge separation in HPP (Supplementary Fig. 7). It is found that such results are comparable to the recently reported analogous polymer photocatalyst[38]. The comparison of visible light driven $CO_2$ reduction to that under UV-visible light is presented in Supplementary Fig. 8. Thus, the photocatalytic $CO_2$ reduction reaction over Pd-HPP-$TiO_2$ depended on UV light of UV-visible light irradiation. The visible light is absorbed by HPP, and most of photons absorbed are changed to heat. When $TiO_2$ is irradiated with UV light to generate electrons and holes in conduction band and valence band, respectively, the electrons transfer at the heterointerface to HPP with π-conjugated structure and can be trapped at Pd(II). It is well know that metals work as electron trap sites to enhance the charge separation efficiency[39].

The highest photocurrent of Pd-HPP-$TiO_2$ can be attributed to the photogenerated electrons transferring from $TiO_2$ to Pd-HPP and to be trapped at Pd. The order of gas evolution rates shown in Fig. 2a is consistent with that of the photocurrent in Fig. 2c, suggesting that efficient electron transfer in Pd-HPP is the dominant influence on the photocatalytic activity with pure $CO_2$.

Pd in Pd-HPP increases the charge separation efficiency, while Pd does not response to the selective $CO_2$ reduction in an aerobic environment. In pure $CO_2$, Pd/$TiO_2$ exhibits high activity for $CH_4$ production with a rate of 104 μmol $g^{-1}$ $h^{-1}$ compared to Pd-HPP-$TiO_2$ (48.0 μmol $g^{-1}$ $h^{-1}$), as shown in Fig. 2d. In the case of similar Pd loading, the higher rate over Pd/$TiO_2$ is ascribed to Pd-HPP absorbing light in a part (Supplementary Fig. 6) and thus decreasing absorbed photon numbers by $TiO_2$. Besides, the light-induced electron transfer from $TiO_2$ to Pd in Pd/$TiO_2$ is more efficient compared with that in Pd-HPP-$TiO_2$, in which Pd sites do not directly contact with hollow $TiO_2$. When the reaction proceeded in diluted $CO_2$ (diluted in $N_2$), the $CH_4$ evolution rate over Pd/$TiO_2$ was higher than that over Pd-HPP-$TiO_2$ (Supplementary Fig. 9a), but the difference between them was less with decreasing the $CO_2$ concentration (Supplementary Fig. 9b), due to the enrichment of low $CO_2$ concentration by the abundant micropores of HPP. The activities for two photocatalysts were close each other using the synthetic gas containing 0.03 vol% $CO_2$ (approximate concentration of air, Supplementary Fig. 9b). However, in air, the $CO_2$ reduction was almost completely inhibited over Pd/$TiO_2$, while it still proceeded over Pd-HPP-$TiO_2$ with the evolution rates of 12.2 μmol $g^{-1}$ $h^{-1}$ and 4.9 μmol $g^{-1}$ $h^{-1}$ for $CH_4$ and CO production, respectively (Fig. 2d and Supplementary Table 2). The calculated conversion yields of $CO_2$ over two catalysts in air and a mixture of $CO_2$ and $N_2$ are compared in Fig. 2e. Monitoring of change in $CO_2$ concentration is important in providing direct evidence for the $CO_2$ conversion, but it has been seldom achieved in the literatures because the change is negligibly little in pure $CO_2$. It is noted that the reduction efficiency in the gas mixture of 0.15 vol% $CO_2$ in $N_2$ is close to that in pure $CO_2$, and the change in $CO_2$ concentration is large enough to calculate the conversion yield, as listed in Supplementary Table 3. Pd/$TiO_2$ is more efficient than Pd-HPP-$TiO_2$ in an anaerobic environment, while the reverse results were observed in air; $CO_2$ conversion yields of 12% and 2.7% over Pd-HPP-$TiO_2$ and Pd/$TiO_2$, respectively, after 2 h UV-visible light irradiation. The yield of 12% is the highest among the $CO_2$ conversions in air reported in the literatures (Supplementary Table 2). The difference between the $CO_2$ conversion yields in 0.15 vol% $CO_2$ in $N_2$ and air is resulted from the absence and presence of $O_2$, respectively. Thus, we investigated the effect of $O_2$ concentration on the photocatalytic reaction. As can be seen in Fig. 2f, for Pd/$TiO_2$, the presence of 0.2 vol% $O_2$ suppressed the $CH_4$ evolution rate, and the presence of 5 vol% $O_2$ dropped it steeply to 6% of that in pure $CO_2$[5]. Interestingly, the negative effect of $O_2$ on the $CH_4$ evolution is significantly less over Pd-HPP-$TiO_2$: the presence of 5 vol% $O_2$ decreased it to 46% of that in pure $CO_2$.

**Porosity and gas uptake.** For heterogeneous catalysis, the reaction rate is usually proportional to the surface coverage of reactants on the catalyst, so the $CO_2$ conversion efficiency particularly relies on the $CO_2$ adsorption on the photocatalysts[33,40,41]. The surface properties including porosity as well as the $CO_2$ adsorption capability and selectivity of as-prepared samples were investigated. As shown in Fig. 3a, the $N_2$ adsorption-desorption isotherms of Pd-HPP and Pd-HPP-$TiO_2$ exhibit a steep increase at relative low pressure ($P/P_0 < 0.001$) and an obvious hysteresis at medium pressure, which indicate the existence of abundant micropores and mesopores[42]. This

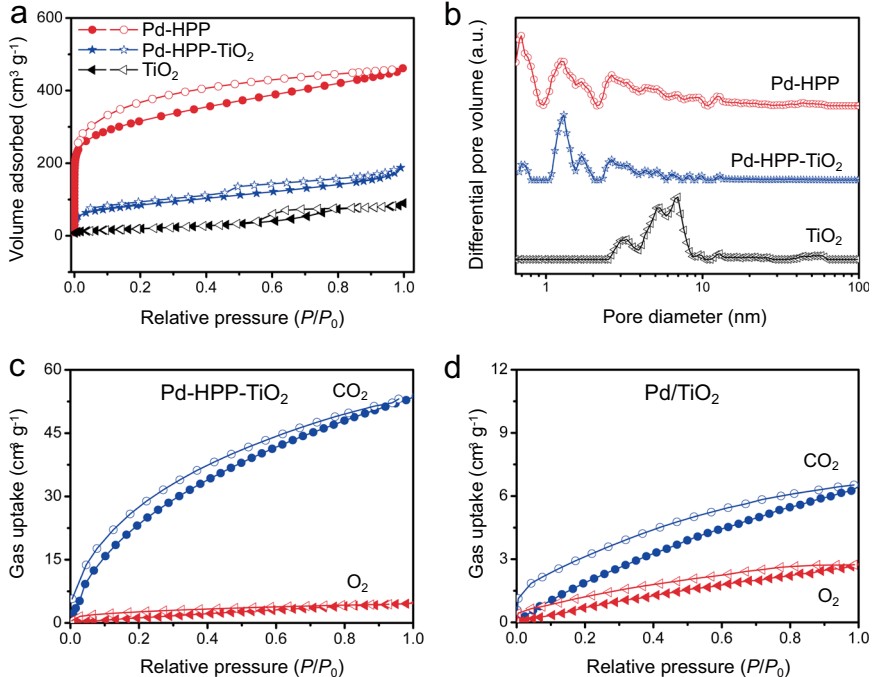

**Fig. 3 Surface porosity and gas uptake.** (**a**) $N_2$ adsorption-desorption isotherms and (**b**) pore size distribution plots of hollow $TiO_2$, Pd-HPP, and Pd-HPP-$TiO_2$. Comparisons in $CO_2$ and $O_2$ uptake of (**c**) Pd-HPP-$TiO_2$ and (**d**) Pd/$TiO_2$ at 273 K.

result may be due to a fast rate of hyper-crosslinking and the low degree of free packing for building blocks by Friedel-Crafts alkylation reaction. In contrast, pure hollow $TiO_2$ shows the character of type IV isotherm with a hysteresis loop at medium pressure, which suggests the formation of mesoporous structure and gives a Brunauer–Emmett–Teller surface area ($S_{BET}$) of 75 m$^2$ g$^{-1}$. Owing to the high microporosity of HPP (0.7 and 1.3 nm), Pd-HPP-$TiO_2$ has a large surface area of 323 m$^2$ g$^{-1}$ and micropore volume of 0.22 cm$^3$ g$^{-1}$ (Supplementary Table 4). The introduction of $TiO_2$ caused a moderate decrease of the ultra-micropore of HPP, while micropores centered at 1.3 nm were largely remained (Fig. 3b). The microporous nature of Pd-HPP-$TiO_2$ causes the $CO_2$ enrichment around the catalytic active Pd sites in Pd-HPP. The $CO_2$ adsorption capability of Pd-HPP-$TiO_2$ reaches as high as 54.0 cm$^3$ g$^{-1}$ at 1.0 bar and 273 K, which is 4.9 times higher than that of $TiO_2$ (Fig. 3c and Supplementary Fig. 10). In contrast, the as-obtained Pd/$TiO_2$ shows a low $CO_2$ uptake of 6.5 cm$^3$ g$^{-1}$ under similar conditions. To study the effect of porphyrin concentration on the adsorption of $CO_2$ and photocatalytic reaction, we have prepared porous Pd-HPP-$TiO_2$ composites with different mass percentage of porphyrin unit by adjusting the adding amount of porphyrin monomer. The $CO_2$ uptake of porous Pd-HPP-$TiO_2$ composites with 53.8 and 74.9 wt% of Pd-HPP was presented in Supplementary Fig. 11. According to the adding amount of porphyrin monomer and the yield of resulted polymer, the mass percentage of porphyrin unit in Pd-HPP is calculated to be about 70.5%. Thus the molar ratios of porphyrin unit/adsorbed $CO_2$ can be calculated and compared in Supplementary Table 5. The results suggest that the ratio of porphyrin/$CO_2$ almost keeps constant. A little lower porphyrin/$CO_2$ ratio in Pd-HPP-$TiO_2$ composites than that in pure Pd-HPP is presumably due to the introduction of $TiO_2$ slightly blocking the crosslinking of porphyrin monomer. It can be concluded that the adsorption of $CO_2$ molecules strongly depends on the porphyrin content. Besides the $CO_2$ adsorption, electron generation on $TiO_2$ photocatalyst and trapping by Pd(II) sites are crucial processes that involved in the photocatalytic reactions. The result of photocatalytic $CO_2$ reduction in Supplementary Fig. 12 reveals that there is an appropriate porphyrin content that balanced the $CO_2$ adsorption and conversion efficiency.

The selectivity ratio of $CO_2$/$O_2$ over Pd/$TiO_2$ is 3.1 calculated by initial slopes of adsorption isotherms in the low pressure region (Fig. 3d and Supplementary Fig. 13), indicating a mediocre $CO_2$ adsorption selectivity in the presence of $O_2$. Interestingly, Pd-HPP-$TiO_2$ exhibits a high $CO_2$/$O_2$ selectivity ratio of 23.9. Moreover, $CO_2$ has delocalized $\pi$-bonds with higher quadrupole moment ($-13.4 \times 10^{-40}$ C m$^2$) than $O_2$ ($-1.03 \times 10^{-40}$ C m$^2$)[43,44]. Introducing porphyrin with a core of four pyrrole rings as the building blocks into microporous materials endows them polarizing N-containing groups and large $\pi$-conjugated structure, which could response to the enhanced interaction with $CO_2$[9–11,45]. The high affinity for $CO_2$ instead of $O_2$ for microporous Pd-HPP-$TiO_2$ is consistent with the selective $CO_2$ adsorption and reduction. Thus the photocatalytic $CO_2$ reduction is achieved in an aerobic environment by taking the advantage of selective $CO_2$ adsorption in microporous HPP.

**Structural analysis**. Structural characterizations provide more information for understanding the selective $CO_2$ adsorption and conversion. Pd-HPP-$TiO_2$ displays the X-ray diffraction (XRD) peaks of pure $TiO_2$ anatase (Fig. 4a). No diffraction peak of Pd crystal indicates Pd (II) coordinates to the porphyrin in Pd-HPP. The observation by scanning electron microscopy (SEM) reveals the morphology of the core-shell $SiO_2$@$TiO_2$ to have a uniform size of 100~200 nm for template-assisted knitting of TPP (Supplementary Fig. 14). The transmission electron microscopy (TEM) and high-resolution TEM (HRTEM) images indicate that hollow $TiO_2$ has the characteristic lattice plane of anatase $TiO_2$ (101), coated by Pd-HPP (Fig. 4b–d and Supplementary Fig. 15). The structure of Pd-HPP-$TiO_2$ was observed by TEM, and no Pd nanoparticle was detected in the HRTEM image, which is consistent with the XRD analysis. As further evidence, the elemental distributions were analyzed by scanning transmission electron microscopy (STEM) and energy-dispersive X-ray (EDX) mapping

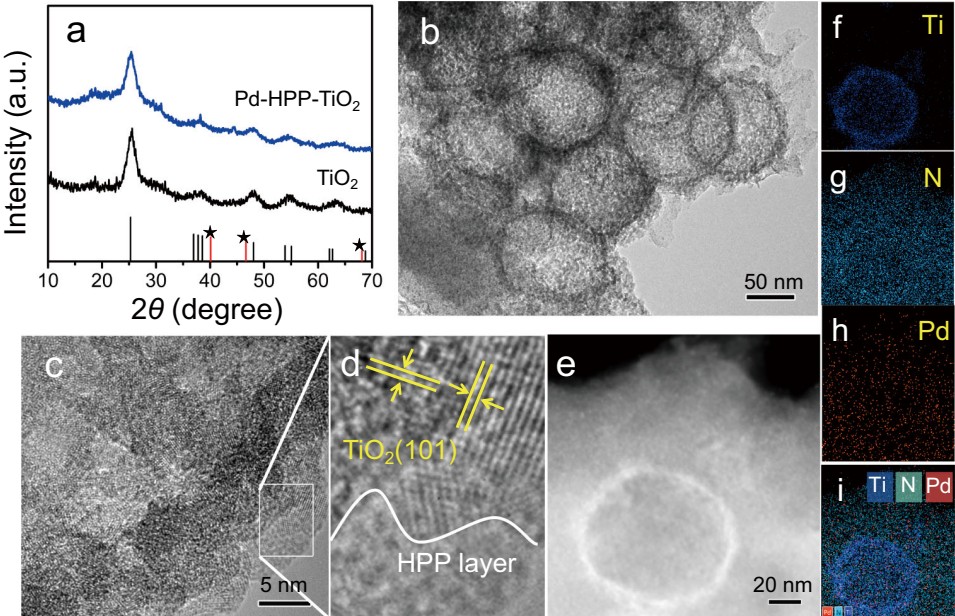

**Fig. 4 Crystal and morphology characterizations. a** XRD patterns of $TiO_2$ and Pd-HPP-$TiO_2$. The vertical lines are the position and intensity of anatase $TiO_2$ (JCPDS 21-1272) and Pd (labeled by the star, JCPDS 46-1043). **b** TEM, **c**, **d** HRTEM, **e** STEM, and **f–i** the corresponding EDX element mapping images of Pd-HPP-$TiO_2$.

tests. Figure 4e–i display that hollow $TiO_2$ is embedded in Pd-HPP with a homogeneous distribution of C, N, and Pd elements.

The chemical structure of Pd-HPP-$TiO_2$ was investigated by Fourier transform infrared (FT-IR) absorption, solid-state $^{13}$C cross-polarization magic-angle spinning nuclear magnetic resonance (CP-MAS NMR), and X-ray photoelectron spectroscopy (XPS) measurements. As shown in FT-IR spectrum, the C-H stretching band at 2920–2960 $cm^{-1}$ indicates the methylene linkage of HPP by solvent knitting (Supplementary Fig. 16)[46]. The broad bands of Ti-O-Ti stretching vibrations at 400–1000 $cm^{-1}$ are observed for $TiO_2$ and Pd-HPP-$TiO_2$. The CP-MAS NMR spectrum of Pd-HPP-$TiO_2$ indicates evidence for the hyper-crosslinking process at the molecular level. The resonance peaks at 128, 137, and 146 ppm are attributed to the carbon atoms in the benzene ring and porphyrin ring (Fig. 5a). The peak with a chemical shift at 37 ppm corresponds to the methylene linkers, indicating the successful linking of TPP via the Friedel-Crafts reaction[47]. The chemical states of elements were analyzed by the XPS spectrum to display the presence of C, N, Ti, O, and Pd in the corresponding samples and the coexistence of them in Pd-HPP-$TiO_2$ (Supplementary Fig. 17). The high-resolution Pd 3$d$ spectra in Fig. 5b show distinct doublet peaks at 343.3 and 338.1 eV, assigned to 3$d_{5/2}$ and 3$d_{3/2}$ of the coordinated Pd(II)[48,49]. In contrast, metallic Pd is predominantly observed in Pd/$TiO_2$ XPS, together with a weak shoulder peak of the adsorbed $Pd^{2+}$ that remained without reducing[49,50]. Combined with the Pd-N signal in N 1$s$ spectrum of Pd-HPP-$TiO_2$ (Supplementary Fig. 17), it is deduced that Pd coordinates successfully with the core of the porphyrin unit as Pd(II) but not as free $Pd^{2+}$ or metallic Pd. The formation of Pd-HPP on hollow $TiO_2$ caused the binding energy of Ti 2$p$ shifting to 0.7-eV higher energy (Supplementary Fig. 17). This shift reflects that the electron density of Ti is decreased by the electronic interaction with Pd-HPP, which is favorable for the electron transfer from $TiO_2$ to Pd-HPP during the photocatalytic reactions under UV-visible light irradiation.

Synchrotron-based X-ray absorption spectroscopy was employed to provide further information on the valence state. Figure 5c shows the Pd K-edge X-ray absorption near-edge structure (XANES) spectra. The absorption edge energy of Pd-HPP-$TiO_2$ is close to that of PdO but higher than that of Pd foil, confirming Pd(II) in Pd-HPP-$TiO_2$. Fourier transform of the extended X-ray absorption fine structure (EXAFS) displays the main peak at 1.5 Å for Pd-HPP-$TiO_2$ (Fig. 5d), arising from Pd-N bonding. No obvious peak was observed at the Pd-Pd position (2.5 Å) of Pd foil, indicating that Pd(II) sites were dispersed in Pd-HPP-$TiO_2$[48,51]. The structural parameters were obtained by the quantitative EXAFS curve fitting (Fig. 5e). Supplementary Table 6 reveals that the coordination number of Pd in Pd-HPP-$TiO_2$ is close to 4.0, indicating that Pd(II) coordinates to four N atoms of the porphyrin. The measured Pd-N bond distance of 2.03 Å is also close to the reported results of Pd-$N_4$ center[48,49]. Inductively coupled plasma mass spectrometry (ICP-MS) provides the accurate element composition, showing that the weight ratios of $TiO_2$ and Pd were 34.4% and 2.72%, respectively (Supplementary Table 7). Meanwhile, the similar Pd content was ensured in the control (Pd/$TiO_2$) to avoid the effect of metal amount on the photocatalytic activity. Based on the above characterizations, it is concluded that HPP is successfully formed on the surface of hollow $TiO_2$ and then Pd(II) coordinates to the porphyrin core of HPP to form Pd-HPP-$TiO_2$.

**Photocatalytic mechanism of $CO_2$ reduction with $H_2O$.** To clarify the reaction pathway, the reaction intermediates of $CO_2$ adsorption and photocatalytic reduction on the surface of Pd-HPP-$TiO_2$ were monitored by in-situ diffuse reflectance infrared Fourier transform spectra (DRIFTS). The spectra demonstrate the adsorption of $CO_2$ and $H_2O$ on Pd-HPP-$TiO_2$ in the dark. The absorption band in the range of 3500–3800 $cm^{-1}$ are in good agreement with those assigned to the stretching vibrations of surface-bonded OH groups and $H_2O$, suggesting the $H_2O$ adsorption on the catalyst surface (Supplementary Fig. 18)[52,53]. The peaks at 1740, 1690, and 1640 $cm^{-1}$ can be assigned to the surface adsorbed carbonate species (Fig. 5f)[54,55]. Under UV-visible light irradiation, the peaks at 1690 and 1640 $cm^{-1}$ were significantly weakened. Meanwhile, the peak at 1740 $cm^{-1}$ first became flat at 2 min and then changed to a negative peak with prolonged irradiation, which could be explained by the existence of pre-adsorbed carbonate species on the Pd-HPP-$TiO_2$ surface

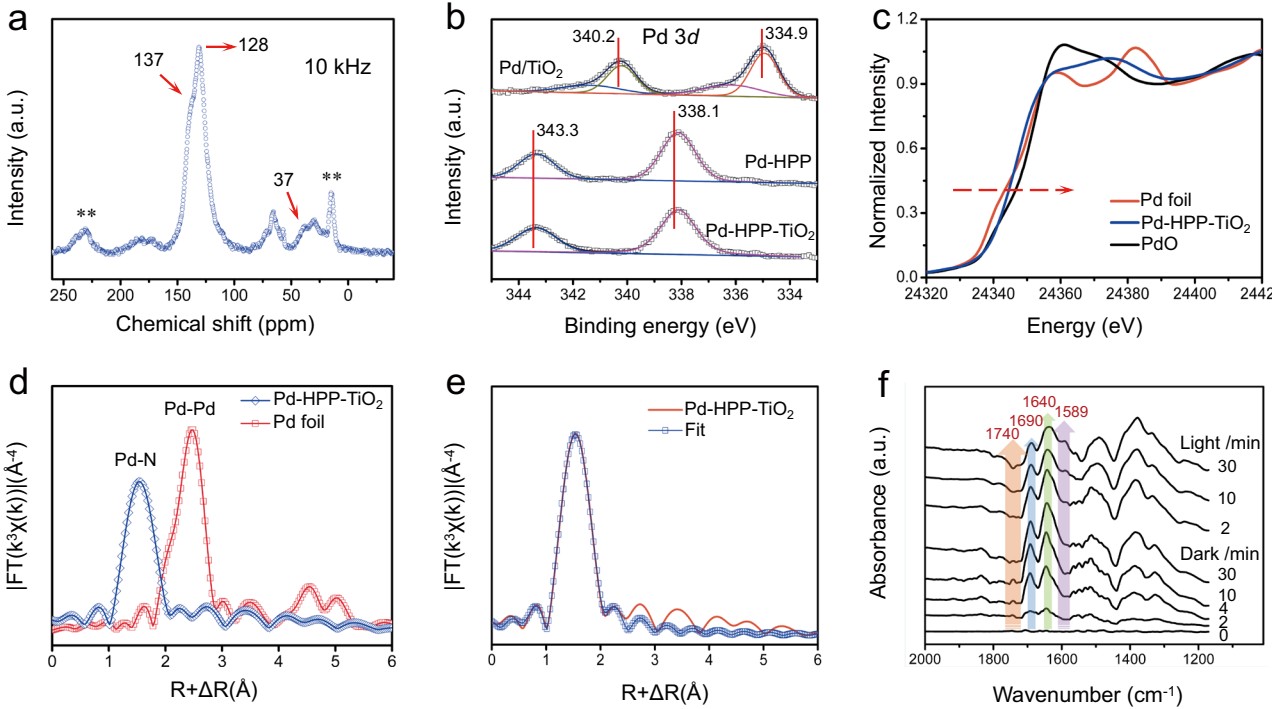

**Fig. 5 Chemical structure analysis and surface species evolution of Pd-HPP-TiO₂. a** Solid-state $^{13}C$ CP-MAS NMR spectrum. **b** Comparison of Pd $3d$ XPS spectra for three photocatalysts. **c** Pd K-edge XANES and **d** Fourier transformed EXAFS spectra of Pd-HPP-TiO₂ and references. **e** Fourier transformed EXAFS spectrum of Pd-HPP-TiO₂ and fitting curve. **f** In situ DRIFTS test of gas adsorption on Pd-HPP-TiO₂ in the dark and during the photocatalytic $CO_2$ reduction under UV-visible light irradiation.

before collecting the baseline due to its high $CO_2$ uptake. The results indicate the efficient consumption of surface carbonate during the photocatalytic reaction. Meanwhile, a new peak at 1589 cm$^{-1}$ emerged in the spectra is suggested to be the C=O stretching vibration of *COOH groups, which was the vital intermediate for *CO formation and then transformed to CO and other fuels.[56–58] According to the detailed studies on the mechanism of $CO_2$ reduction, there are two possible pathways, i.e., one is the formaldehyde pathway and the other is the carbene pathway[59–61]. Although formaldehyde and methanol have been reported as products in some setups, they are not detected in this work. The photocatalytic $CO_2$ reduction in the gas-solid reaction can normally form CO and CH₄.[25,26] Therefore, the $CO_2$ reduction is more likely to be a carbene pathway as $CO_2 \rightarrow COOH \rightarrow CO \rightarrow \bullet C \rightarrow \bullet CH_3 \rightarrow CH_4$. The $CO_2$ molecules are activated at Pd(II) sites in Pd-HPP and then react with the proton and electron to produce the intermediate *COOH. Subsequent reaction with the electron and proton results in the splitting of *COOH into *CO and H₂O. At this time, part of CO is produced by *CO desorption. Since Pd can also function efficiently for the decomposition of H₂O to generate Pd-H species[62,63], the adsorbed *CO on Pd will be further react with the dissociated H to the formation of •C radicals, which can successively combine with •H radicals, thereby forming •CH, •CH₂, •CH₃, and finally CH₄ product[53,64–66].

The cycling experiment and related characterizations were carried out to investigate the photocatalytic sites of CH₄ evolution and the stability of Pd-HPP-TiO₂. As shown in Supplementary Fig. 19, Pd-HPP-TiO₂ showed about 80% of the initial photocatalytic activity for the $CO_2$ conversion after five consecutive cycles. The thermal stability of HPP is maintained up to 330 °C, as confirmed by thermogravimetric analysis (Supplementary Fig. 20). The comparison in FT-IR spectra of Pd-HPP-TiO₂ before and after the photocatalytic reaction reveals no obvious change in the chemical structure of Pd-HPP

(Supplementary Fig. 21). A new peak at 40.1° is observed in the XRD pattern of Pd-HPP-TiO₂ after the cycling test to be assigned to Pd(0) (Supplementary Fig. 22), implying that a fraction of the coordinated Pd(II) was reduced to Pd(0) dissociating from the coordination sites to form Pd nanoparticles. XPS analysis in Supplementary Fig. 23a corroborates the reduction, leading to a decrease of the photocatalytic activity during the recycle. Such reduction suggests the electron trapped at Pd(II) for $CO_2$ reduction[67]. After five cycles, ~79% Pd existed as Pd(II) owing to the coordination to the porphyrin in microporous HPP, indicating the considerable stability of Pd-HPP-TiO₂ structure during the photocatalytic reaction. Pd nanoparticles are less active for the photocatalytic $CO_2$ reduction in air (Fig. 2e), compared to Pd(II) in Pd-HPP-TiO₂. Therefore, a tentative mechanism of photocatalytic $CO_2$ reduction over Pd-HPP-TiO₂ is proposed in Fig. 1. Benefiting from the high $CO_2$ adsorption capability and selectivity of HPP, $CO_2$ can be selectively enriched in Pd-HPP. Under UV light irradiation, the photogenerated electrons in the conduction band of TiO₂ transfer efficiently to Pd-HPP, the electrons are trapped at the coordinated Pd(II) in Pd-HPP, and the adsorbed $CO_2$ on Pd(II) is reduced to produce CH₄ and CO accompanied by the recovery of Pd(II), while the holes in the valence band of TiO₂ can oxidize water that adsorbed on hollow TiO₂ to produce O₂. The simultaneous monitoring of concentrations of $CO_2$ (decreased), CH₄, CO, and O₂ (produced), corroborates the overall redox reaction over Pd-HPP-TiO₂, applicable to aerobic environment, especially for flue gas with 3~5 vol% O₂ and air with 300~400 ppm of $CO_2$ and ~21 vol% O₂ content.

In summary, we have demonstrated that Pd-HPP-TiO₂, constructed based on higher $CO_2$ adsorption capability than O₂ and efficient charge separation for $CO_2$ reduction and H₂O oxidation, exhibits high photocatalytic activity for $CO_2$ reduction in an aerobic environment. In the presence of 5 vol% O₂, the $CO_2$ reduction over a catalyst without HPP (Pd/TiO₂) steeply drops to 6% of that in pure $CO_2$. In contrast, the O₂ inhibition is significantly less over Pd-HPP-

$TiO_2$, which maintained 46% of the $CH_4$ evolution rate in pure $CO_2$. Pd-HPP-$TiO_2$ shows the photocatalytic activity even in air with the $CO_2$ conversion yield of 12% and the $CH_4$ production of 24.3 μmol $g^{-1}$ after 2 h UV-visible light irradiation, 4.5 times higher than those over Pd/$TiO_2$. The HPP layer effectively enriches $CO_2$ at Pd(II) to lessen the $O_2$ reduction. Water adsorbed on $TiO_2$ is oxidized by the holes in the valence band of $TiO_2$, leading to reduce the charge recombination and enhance $CO_2$ conversion. This study presents an insight into realizing the photocatalytic selective $CO_2$ reduction for effectively reducing $CO_2$ concentration in air or flue gas and producing valuable solar fuels as well.

## Methods

**Preparation of core-shell $SiO_2$@$TiO_2$ and hollow $TiO_2$ sphere**. The preparation of $SiO_2$@$TiO_2$ was referenced in the literature[68]. In a 100 mL round bottom flask, a mixture containing 79 mL of ethanol, 3.9 mL of ammonia solution, and 1.4 mL of water was mixed with 1.0 g of $SiO_2$ nanoparticles with diameter of about 100 nm to obtain a $SiO_2$ colloidal solution. Then, 28 mL of acetonitrile was added to the above mixture with stirring at 4 °C. A solution containing 36 mL of ethanol, 12 mL of acetonitrile, and 1 mL of titanium isoporpoxide was added dropwise to the colloidal $SiO_2$ solution. The mixture was stirred vigorously for 12 h, and the resulting white solution was dried in an oven at 80 °C. After calcining the solution at 600 °C for 6 h, core-shell $SiO_2$@$TiO_2$ with diameter of 100–150 nm was obtained as a white powder. Hollow $TiO_2$ with thickness of about 10 nm were prepared by etching $SiO_2$@$TiO_2$ in 10 mL of 2.5 M NaOH solution for 2 days.

**Preparation of Pd-HPP-$TiO_2$**. HPP-$TiO_2$ was synthesized by an in-situ knitting method using $SiO_2$@$TiO_2$, 5,10,15,20-tetraphenylporphyrin (TPP), and dichloromethane (DCM, 8 mL) as a solid template, building block, and solvent, respectively. After uniform dispersion of $SiO_2$@$TiO_2$ and TPP in DCM, $AlCl_3$ catalyst was added at 0 °C with constant stirring. The reaction mixture was stirred at 0 °C for 4 h, 30 °C for 8 h, 40 °C for 12 h, 60 °C for 12 h, and 80 °C for 24 h under the protection of $N_2$ gas. Then, the sample was filtrated and washed twice with water and twice with ethanol, followed by further purification by extracting with ethanol for 2 days. Finally, the obtained solid was dried in a vacuum drying oven at 60 °C for 24 h, and etched in a 2.5 M NaOH solution for 2 days to yield HPP-$TiO_2$. 40 mg of HPP-$TiO_2$ or HPP was dispersed in 4 mL of acetonitrile and ultra-sounded for 5 min to obtain a homogeneous solution. Then 3 mL of $H_2PdCl_4$ solution (1.08 mg mL$^{-1}$ Pd) was added and kept at 40 °C for 12 h. After the reaction stopped, the product was washed twice with water and twice with acetone, and vacuum dried at 60 °C overnight.

**Preparation of Pd/$TiO_2$ as a reference**. Pd nanoparticles were deposited on the surface of hollow $TiO_2$ from the photoreduction of Pd(II) on $TiO_2$: 100 mg of hollow $TiO_2$ was dispersed in 90 mL of deionized water and 10 mL of methanol. After ultrasonic treatment for 20 min, a certain amount of $H_2PdCl_4$ solution was added. Then, $N_2$ was continuously flowed into the solution for 20 min to ensure $N_2$-saturation. A mercury lamp was used for the photoreduction of $H_2PdCl_4$ on hollow $TiO_2$. After the photocatalytic reaction for 4 h, the solution was centrifuged and washed three times with ethanol and twice with water, and then dried overnight in a vacuum at 60 °C to yield Pd/$TiO_2$ as a gray powder.

**Characterizations**. Gas ($N_2$, $CO_2$, $O_2$) adsorption-desorption isotherms were analyzed by TriStar II 3flex adsorption apparatus (Micromeritics, USA). Samples were degassed at 100 °C for 12 h under vacuum before analysis. The structure and crystallinity of the samples were characterized using X-ray diffraction (XRD) analysis on an X-Pert PRO diffractometer with Cu-Kα radiation. The field emission scanning electron microscopy (FESEM) images were recorded by using a field emission scanning electron microscope (FEI Sirion 200, USA) at 10 kV. The high-resolution transmission electron microscopy (HR-TEM) and scanning transmission electron microscopy (STEM) images of samples were recorded on Tecnai G2 F30 microscope (FEI, Holland). The FT-IR experiment was conducted on a Bruker ALPHA Fourier transform infrared spectrometer. The Solid-state $^{13}C$ CP/MAS NMR spectra were tested on a WB 400 MHz Bruker Avance II spectrometer. The $^{13}C$ CP/MAS NMR spectra were collected on a 4 mm double resonance MAS probe at a rotation rate of 10 kHz. X-ray photoelectron spectroscopy (XPS) was measured with a monochromatic Mg Kα source on Thermo VG scientific ESCA MultiLab-2000, and the data were calibrated according to the C (carbon) 1 s peak (binding energy = 284.6 eV). The X-ray absorption spectra were collected on the beamline BL01C1 in NSRRC, and were provided technical support by Ceshigo Research Service "www.ceshigo.com". The radiation was monochromatized by a Si (111) double-crystal monochromator. XANES and EXAFS data reduction and analysis were processed by Athena software. The actual contents of Ti and different metals were measured by inductively coupled plasma mass spectrometry (NexION 300X, Perkin Elmer, USA). Photoluminescence (PL) emission spectra were collected by a Hitachi F-7000 spectrofluorometer at the excitation wavelength of 360 nm. UV-vis diffuse reflectance spectra (DRS) were obtained using a UV-vis spectrophotometer (UV-3600, Shimadzu, Japan). The reduction products from $^{13}CO_2$ were analyzed by HP 5973 gas

chromatography-mass spectrometry (GC-MS). Thermogravimetric analysis (TGA) was carried out in $N_2$ and air from room temperature to 850 °C using the Perkin Elmer instrument Pyris1 TGA with a heating rate of 10 °C min$^{-1}$. The electrochemical and photoelectrochemical properties of the sample were tested using an electrochemical workstation (CHI650E, Chenhua Com., China) with a standard three-electrode system. A Pt wire and Ag/AgCl were used as the counter and reference electrodes, respectively. 5 mg of a catalyst was dispersed into 1 mL of 1:1 isopropanol/$H_2O$ containing 10 μL of Nafion. Then, 50 μL of the above suspension was coated on an ITO glass as a working electrode. Electrochemical impedance spectra (EIS) were obtained in 0.1 M KCl electrolyte containing 5 mM Fe(CN)$_6$$^{3-}$/Fe(CN)$_6$$^{4-}$. Photocurrent signals were detected in 1 M $Na_2SO_4$ solution during light-on and light-off cycles.

**Photocatalytic $CO_2$ reduction**. 30 mg of powder sample was dispersed on the middle of the culture ware, placed in the sealed custom-made glass vessel, 10 cm away from the light source. The photocatalytic $CO_2$ reduction reaction was performed under 1 atm of a certain atmosphere (pure $CO_2$, $CO_2$/$O_2$ mixed gas, or air). 2 mL of water, as proton source for the $CH_4$ production, was dropped to the bottom of glass vessel and vaporized on standing. The reaction mixture was irradiated using a 300 W Xenon lamp (PLS-SXE300D, Beijing Perfectlight Technology, China) as light source. The full spectrum locates in the UV-visible light region (325~780 nm), as shown in Supplementary Fig. 24. For the visible light driven experiment, a cut-off filter of 420 nm was equipped with the lamp. The generated gases were analyzed by a gas chromatography analyzer (Shimadzu GC-2014C, Japan) with a flame ionization detector (FID) and an optic fiber oxygen sensor (Ocean-Optics, UK). The yield of $CO_2$ conversion was conducted in 0.2 vol% $CO_2$ in $N_2$ or air under irradiation for 4 h. To exclude the self-decomposition of samples, the sample was firstly irradiated for 4 h under $N_2$ to confirm the stability.

## Data availability

The data supporting the findings of this study are available in the paper and Supplementary Information. Source data are provided with this paper.

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

## Acknowledgements

This work was financially supported by the National Natural Science Foundation of China (22122602, 21771070, 22161142005) and the Fundamental Research Funds for the Central Universities (2019KFYRCPY104, 2018KFYYXJJ120). We thank the Analysis and Testing Center, Huazhong University of Science and Technology for the characterization of materials.

## Author contributions

J.W. conceived the project and designed the experiments. Y.M. and X.Y. performed the experiments and analyzed the data. S.W. helped the synthesis. T.L. and B.T. discussed the

results. C.C. designed the photocatalytic tests. B.T., C.C., T.M., E.R.W. and H.Z. helped in improving the manuscript. Y.M., J.W., and H.Z. co-wrote the manuscript.

## Materials availability

## Competing interests

The authors declare no competing interests.
