## [Peer Review File · Nature Communications]

Selective photocatalytic CO₂ reduction in aerobic environment by microporous Pd-porphyrin-based polymers coated hollow TiO₂REVIEWER COMMENTS

Reviewer #1 (Remarks to the Author):

This MS deals with the photocatalytic reduction of CO₂ with H₂O on Pd-porphyrin-based polymer coated hollow TiO₂ by applying various spectroscopic measurements and ¹³CO₂ tracer and reported some interesting results and discussions. However, the following points should be clarified and described clearly before the publication in Nature Communication since their discussions are rough.

- 1) At page 3, in Introduction, the authors described that “The heteroatoms-rich microporous structure can not only improve the CO₂ adsorption capability and selectivity but also stabilize-Pd(II) sites,”. What does it mean “selectivity”? This should be clearly described.
- 2) At page 4, the authors utilized isotopically labelled ¹³CO₂ to check the source of the products. CH₄ is confirmed to be ¹³CH₄. However, they didn't show the origin of the formation of CO. This point should also be clarified.
- 3) In addition to the above point using labelled ¹³CO₂, the authors should show the results obtained by infrared Fourier transform spectra when they used ¹³CO₂ instead of CO₂ since it is not easy to assign the spectra at around 3627 cm⁻¹ due to HCO₃⁻, at around 1740 cm⁻¹ due to CO₃⁻, nor at around 1589 cm⁻¹ due to CO stretching. They should show the results obtained using ¹³CO₂ in those IR spectra.
- 4) At page 5, the authors mentioned that “Photoluminescence of Pd-HPP-TiO₂ was almost quenched, indicating that the recombination of the photo-generated electrons and holes was effectively reduced by efficient charge separation.” However, the authors' explanation is not correct and wrong since there are many processes to lead an efficient quenching of photoluminescence due to a radiative recombination of the photo-generated electrons and holes in TiO₂ (see in Adv. Catal. 44, 119-257 (2000) and references therein.). Therefore, above sentences should be reconsidered and corrected in a right description.
- 5) At page 7, the authors mentioned that “... the existence of abundant micropores, and ... the existence of mesopores.” However, what is the origin to produce both micro- and meso-pore in hollow TiO₂? This point should be described.
- 6) At page 8, the authors described that “Introducing porphyrin with a core of four pyrrole rings as the building blocks into microporous materials endows them polarizing N-containing groups and large π-conjugated structure, which could response to enhance the interaction with CO₂.” However, in order to understand these sentences, the readers need, at least, the following data. What is the ratio of porphyrin/CO₂ in their system? Also, they should show the effect of porphyrin concentration on the adsorption of CO₂ and photocatalytic reaction. These are very important points to accept their arguments.
- 7) At page 9, the authors mentioned that “... the coordination number of Pd in Pd-HPP-TiO₂ is close to 4.0,” However, in Pd-K-edge XANES spectrum (Fig. 5(c)), we cannot see any pre-edge peak due to the 4 –

coordinated species. What is the reason? Is it really 4-coordinated species but not 6-coordinated species?

8) What is the bond distance of Pd-N? It should be obtained from FT-EXAFS spectrum.

9) At page 11, as the authors described, Pd(II) species slowly changes to Pd(0) under UV-visible light irradiation. In fact, as shown in Fig. S15, the formation of CH₄ and CO decrease with increasing irradiation time and also repeating the cycles. Despite these results, the authors mentioned that “indicating the durability of Pd-HPP-TiO₂ for a long-term photocatalytic reaction.” These description is a little rough and rather wrong. This point should be revised correctly, reflecting the real results.

10) Regarding the reaction mechanism in Fig. 1, the authors should be discuss the details on the following points. For the formation of CH₄, it is necessary to be formed the bare C from CO₂ via CO. The mechanism for the formation of bare C should be discussed. And also, for the formation of CH₄, H atoms are required. This point should also be discussed.

Reviewer #2 (Remarks to the Author):

The manuscript by Ma et al. describes the reduction of CO₂ in air to CO and CH₄ over a Pd-HPP-TiO₂ catalyst. The direct conversion of CO₂ in flue gas or air to fuels is vital. However, the reduction of CO₂ to C₁ products have already been extensively studied and the C₁ yield of this work is low. Besides, the catalyst material is too complicated. The functions of different parts of the composite: Pd, HPP and TiO₂, and their synergistic interactions are not clearly explained. Therefore, the manuscript is ill-suited to be published in Nature Communications. Other comments are listed below:

1. The evolution rates for CO and CH₄ are relatively low and the reduction of CO₂ is not selective to a single product.
2. Some language errors should be corrected in this manuscript, such as “followed by loading Pd(II) to via coordination with HPP” in the introduction section.
3. In the introduction section, “Pd allows us to confirm the influence on the reduction.” What influence? the authors should clarify the meanings of the sentence.
4. As shown in Supplementary Fig. S5, after the addition of HPP, the composite showed favourable absorption in the visible light region. Why are not the CO₂ photoreduction experiments carried out under visible light irradiation?

5. The CO₂ photoreduction testing time is too short (only 4 h). And the XRD pattern of Pd-HPP-TiO₂ after the cycling test was changed, indicating that the stability of the catalyst is unsatisfactory.

Reviewer #3 (Remarks to the Author):

The authors report on the preparation, characterization and testing of a Pd-HPP-TiO₂ photocatalyst, that enables effective CO₂ to methane conversion, even in the presence of oxygen gas, due to a preferential adsorption of CO₂ over O₂ in the material. The concept of developing O₂-tolerant CO₂ conversion photocatalysts is noteworthy, considering applications in e.g. flue gas treatment, and is thus expected to attract attention from a broad audience. The work in itself is quite solid and very complete. All proper benchmark experiments have been carried out (e.g. dark reference, Pd directly on TiO₂, Pd-HPP alone, etc.). The preferential and quantitative adsorption of CO₂ on the Pd-HPP-TiO₂ catalyst is demonstrated well, as well as its chemical structure (i.e. indeed Pd(II) instead of Pd(0) NPs, coupled to the polymer). Also, the improved charge transfer dynamics have been characterized appropriately using electrochemical tools. Hence, I recommend publication of this manuscript, considering only a few minor comments:

- 1) at the bottom of page 2 the authors mention an anaerobic environment, while I believe it should be aerobic
- 2) Only a Xe lamp is mentioned in the method section as the light source, while the text also mentions AM 1.5 simulated solar light. Have any filters been used? How is the absolute irradiance at sample distance controlled and experimentally verified?
- 3) The CO₂ conversion selectivity of photocatalysts containing Pd-species towards methane could be further corroborated with recent literature (e.g. Chemical Engineering Journal 410 (2021) 128234)
- 4) On p. 6 it is hypothesized that visible light absorption is mainly converted into heat, while UV absorption by TiO₂ truly results in the observed activity after charge transfer. While I, too, believe this is most probable, has it actually been verified experimentally? For instance, a UV cut-off filter at 420 nm could be used to verify the absence/presence of purely visible light photoactivity?
- 5) On page 7 a type I isotherm is mentioned, which does not correspond with the fact that a H4 type hysteresis loop is observed (characteristic for hollow spheres with walls composed of ordered mesoporous material). It is therefore rather a type IV isotherm.
- 6) It would be good to improve the overall level of English, as the text contains quite some (grammatical) errors, which tend to distract the reader's attention from the actual nice experimental work.

Given below are our responses (in BLUE colour) to the reviewer' comments. The changes to the manuscript and supplementary information are marked in RED colour.

Reviewer #1

This MS deals with the photocatalytic reduction of CO₂ with H₂O on Pd-porphyrin-based polymer coated hollow TiO₂ by applying various spectroscopic measurements and ¹³CO₂ tracer and reported some interesting results and discussions. However, the following points should be clarified and described clearly before the publication in Nature Communication since their discussions are rough.

Response: We thank the reviewer for appreciating the idea of work presented in this manuscript. We also appreciate the reviewers' comments and suggestions that helped us to significantly improve the manuscript. We have addressed the reviewers' comments point-wise in our response below:

1. At page 3, in Introduction, the authors described that “The heteroatoms-rich microporous structure can not only improve the CO₂ adsorption capability and selectivity but also stabilize-Pd (II) sites,”. What does it mean “selectivity”? This should be clearly described.

Response: Thanks for your suggestion. We would like to clarify the meaning of “selectivity” here, i.e., the CO₂ adsorption against O₂ adsorption. In air and flue gas, the CO₂ adsorption and activation on the surface of photocatalysts are low, due to the competitive O₂ adsorption and reduction, as well as the low CO₂ concentration. Therefore, to control CO₂ emission from exhaust gas and reduce CO₂ concentration in air, developing efficient photocatalysts with selective CO₂ adsorption and conversion in an aerobic environment remains a challenge. To address the challenge, we presented a porous composite structure of Pd-porphyrin-based polymer coated hollow TiO₂. High CO₂ adsorption capability and selectivity of porphyrin-based microporous polymers with heterocyclic skeleton and large π -conjugated structure can enrich the low

concentration of CO₂ on the catalytic active sites for reduction without separation from aerobic mixtures.

To make the meaning of this sentence more clearly, we have revised it on Page 3 “The heteroatom-rich microporous structure can not only improve the capability and selectivity of CO₂ adsorption in an aerobic environment but also stabilize Pd(II) sites”.

2. At page 4, the authors utilized isotopically labelled ¹³CO₂ to check the source of the products. CH₄ is confirmed to be ¹³CH₄. However, they didn't show the origin of the formation of CO. This point should also be clarified.

Response: As inspired by this suggestion, we tried our best to study the origin of CO product. When measuring the GC-MS of gas products, a HP-PLOT/Q chromatographic column was previously used due to its superior separation efficiency of CH₄, while it could not separate CO from the air, so the signal of CO was overlapped by that of air. To separate CO from the gas mixture, we have also equipped a chromatographic column of CP-Molsieve 5A with the GC-MS instrument. In this way, the ion fragment analysis of CO product can be realized. It is noted that very few literatures realized the separation of both CH₄ and CO by GC-MS. The result has been added in Supplementary Fig. 3, which confirmed the formation of CO from CO₂ reduction.

We have now also included the corresponding description on Page 4 in the manuscript. “Isotopically labelled ¹³CO₂ (¹³C enrichment of ≥ 97%) was used as the reactant to study the origin of products. According to the ion fragment analysis, the peaks at 2.5 min and 7.2 min in the gas chromatography could be assigned to CH₄ and CO, respectively. As compared to the signals of products under ¹²CO₂, the appearance of ion fragment peaks at m/z = 17 and 29 reveals that the produced ¹³CH₄ and ¹³CO originated from ¹³CO₂ reduction over Pd-HPP-TiO₂ (Supplementary Fig. 3).”

Supplementary Figure S3. (c) GC charts and (d) MS of CO produced from the isotopically labelled $^{13}\text{CO}_2$ (^{13}C enrichment of $\geq 97\%$) used as the reactant.

3. In addition to the above point using labelled $^{13}\text{CO}_2$, the authors should show the results obtained by infrared Fourier transform spectra when they used $^{13}\text{CO}_2$ instead of CO_2 since it is not easy to assign the spectra at around 3627 cm^{-1} due to HCO_3^- , at around 1740 cm^{-1} due to CO_3^- , nor at around 1589 cm^{-1} due to CO stretching. They should show the results obtained using $^{13}\text{CO}_2$ in those IR spectra.

Response: As suggested, we have measured the in-situ DRIFTS spectra in $^{13}\text{CO}_2$ environment. It is found that the peaks of OH stretching (3729 , 3699 , 3627 and 3596 cm^{-1}) are similar to those in $^{12}\text{CO}_2$. Meanwhile, there are no obvious changes in them during light irradiation. Therefore, the absorption band in the range of $3500\text{-}3800\text{ cm}^{-1}$ has nothing to do with CO_2 adsorption, but comes from the stretching vibrations of surface-bonded OH groups and H_2O , suggesting the H_2O adsorption on the catalyst surface (Supplementary Fig. 18). The peaks at 1740 , 1690 and 1640 cm^{-1} can be assigned to surface adsorbed carbonate CO_3^{2-} species (Fig. 5f). In $^{13}\text{CO}_2$ isotopic experiment, these bands exhibit shifts to 1740 , 1690 and 1640 cm^{-1} , as shown in Fig. R1. Although there are somewhat shifts between $^{12}\text{CO}_2$ and $^{13}\text{CO}_2$, the differences in Fig. R1 are not as significant as the literature reports (*J. Mater. Chem. A* 2021, 9, 4291; *J. Am. Chem. Soc.* 2018, 140, 4363; *J. Am. Chem. Soc.* 2016, 138, 9959). Actually, we have repeated this isotopic experiment for five times and found the measurement cannot ensure the pure $^{13}\text{CO}_2$ environment due to the unavoidably residual air in the optical

system as well as the high capability of pre-adsorbed $^{12}\text{CO}_2$ on Pd-HPP-TiO₂. Besides, such experiment is very expensive due to the requirement of continuous $^{13}\text{CO}_2$ gas flow. To the best of our knowledge, most literatures did not realize the comparison of adsorbed $^{12}\text{CO}_2$ and $^{13}\text{CO}_2$ on photocatalyst surface by in-situ DRIFTS measurement. In the above-mentioned literature, they sampled the solution (liquid reaction) for IR measurement after bubbling of $^{12}\text{CO}_2$ or $^{13}\text{CO}_2$ and observed the shift of the carbonate absorption from 1640 to 1590 cm^{-1} (*J. Am. Chem. Soc.* 2016, 138, 9959). In another literature, they used a custom-made Fourier-transform infrared spectrometer with ultrahigh vacuum and a gas IR cell (*J. Mater. Chem. A* 2021, 9, 4291). They directly measured the gas products by the gas headspace injection and the signals were magnified by the repeated reflection in the custom-made gas IR cell (32 scans with a resolution of 0.5 cm^{-1}). This strategy for measuring the gas product by IR is similar to that by GC-MS. It is noted that the isotopic experiment by GC-MS are absent in these literatures. Therefore, in our future work, we may customize such an IR equipment to study the similarity and difference with GC-MS.

According to the suggestion, we have revised the corresponding description on Page 11 in the manuscript. “The absorption band in the range of 3500-3800 cm^{-1} are in good agreement with those assigned to the stretching vibrations of surface-bonded OH groups and H₂O, suggesting the H₂O adsorption on the catalyst surface (Supplementary Fig. 18)^{52,53}. The peaks at 1740, 1690 and 1640 cm^{-1} can be assigned to the surface adsorbed carbonate species (Fig. 5f)^{54,55}. Under UV-visible light irradiation, the peaks at 1690 and 1640 cm^{-1} were significantly weakened and that at 1740 cm^{-1} changed to a negative peak, indicating the efficient consumption of surface carbonate during the photocatalytic reaction.”

Figure R1. In situ FT-IR spectra of the photocatalytic reduction in $^{12}\text{CO}_2$ and $^{13}\text{CO}_2$ on Pd-HPP-TiO₂ under UV-visible light irradiation.

4. At page 5, the authors mentioned that “Photoluminescence of Pd-HPP-TiO₂ was almost quenched, indicating that the recombination of the photo-generated electrons and holes was effectively reduced by efficient charge separation.” However, the authors’ explanation is not correct and wrong since there are many processes to lead an efficient quenching of photoluminescence due to a radiative recombination of the photo-generated electrons and holes in TiO₂ (see in *Adv. Catal.* 44, 119-257 (2000) and references therein.). Therefore, above sentences should be reconsidered and corrected in a right description

Response: We appreciate the reviewer’s kind suggestion. Photoluminescence of TiO₂ can be defined as the radiation emitted from the transition of electronic excited state to its ground electronic state after it has absorbed energy from an external source. The quenching of photoluminescence of TiO₂ mainly involves nonradiative deactivation and deactivation by quencher molecules. When coating with Pd-HPP, the photoluminescence of TiO₂ was almost quenched, indicating the efficient suppression of photo-generated charge recombination through radiative pathways. The photogenerated electrons are expected to transfer from the photoexcited TiO₂ to Pd(II) sites in HPP, leading to effective separation of electrons from the holes left in TiO₂. In

addition, the introduction of Pd enhanced the interaction with gas molecules such as O₂ and CO₂ from air, which also causes the quenching of photoluminescence on TiO₂ surface (*Adv. Catal.* 2000, 44, 119; *J. Phys. Chem.* 1984, 88, 5556; *J. Phys. Chem.* 1997, 101, 2632.).

We have revised the corresponding description on Page 6 in the manuscript. “When coating with Pd-HPP, the photoluminescence of TiO₂ was almost quenched, indicating the efficient suppression of photo-generated charge recombination through radiative pathways. The photogenerated electrons are expected to transfer from the photoexcited TiO₂ to Pd(II) sites in HPP, leading to effective separation of electrons from the holes left in TiO₂. In addition, the introduction of Pd enhanced the interaction with gas molecules such as O₂ and CO₂ from air, which also causes the quenching of photoluminescence on TiO₂ surface³⁴⁻³⁶.”

34. Anpo, M. et al. Applications of photoluminescence techniques to the characterization of solid surfaces in relation to adsorption, catalysis, and photocatalysis. *Adv. Catal.* **44**, 119–257 (1999).

35. Anpo, M. et al. Photoluminescence of zinc oxide powder as a probe of electron-hole surface processes. *J. Phys. Chem.* **88**, 5556–5560 (1984).

36. Anpo, M. et al. Photocatalytic reduction of CO₂ with H₂O on titanium oxides anchored within micropores of zeolites: effects of the structure of the active sites and the addition of Pt. *J. Phys. Chem. B* **101**, 2632–2636 (1997).

5. At page 7, the authors mentioned that “. . . the existence of abundant micropores, and . . . the existence of mesopores.” However, what is the origin to produce both micro- and meso-pore in hollow TiO₂? This point should be described.

Response: We are sorry for the unclear descriptions of pore structures. Pure hollow TiO₂ possesses a typical mesoporous structure, which is confirmed by a type IV isotherm with a hysteresis loop at medium pressure. Hollow TiO₂ spheres were obtained via a two-step route, which consists of assembling TiO₂ nanoparticles on SiO₂ core template and then etching the SiO₂ core template. Hence the mesopores were produced inside the assembled TiO₂ nanoparticles. The Pd-HPP and Pd-HPP-TiO₂ exhibit a steep

increase at relative low pressure ($P/P_0 < 0.001$) and an obvious hysteresis at medium pressure, which indicate the existence of abundant micropores and mesopores. This result may be due to a fast rate of hyper-crosslinking and the low degree of free packing for building blocks by Friedel-Crafts alkylation reaction.

We have revised them on Page 8 in the revised manuscript. “As shown in Fig. 3a, the N_2 adsorption-desorption isotherms of Pd-HPP and Pd-HPP-TiO₂ exhibit a steep increase at relative low pressure ($P/P_0 < 0.001$) and an obvious hysteresis at medium pressure, which indicate the existence of abundant micropores and mesopores⁴². This result may be due to a fast rate of hyper-crosslinking and the low degree of free packing for building blocks by Friedel-Crafts alkylation reaction. In contrast, pure hollow TiO₂ shows the character of type IV isotherm with a hysteresis loop at medium pressure, which suggests the formation of mesoporous structure and gives a Brunauer–Emmett–Teller surface area (S_{BET}) of $75 \text{ m}^2 \text{ g}^{-1}$.”

6. At page 8, the authors described that “Introducing porphyrin with a core of four pyrrole rings as the building blocks into microporous materials endows them polarizing N-containing groups and large -conjugated structure, which could response to enhance the interaction with CO₂.” However, in order to understand these sentences, the readers need, at least, the following data. What is the ratio of porphyrin/CO₂ in their system? Also, they should show the effect of porphyrin concentration on the adsorption of CO₂ and photocatalytic reaction. These are very important points to accept their arguments.

Response: This suggestion inspired us to study the relationship between porphyrin content and CO₂ adsorption capacity. Pure Pd-HPP possesses a CO₂ uptake of $96 \text{ cm}^3 \text{ g}^{-1}$ under standard conditions. According to the adding amount of porphyrin monomer and the yield of resulted polymer, the mass percentage of porphyrin unit in Pd-HPP is calculated to be about 70.5%, giving an average value of porphyrin unit/adsorbed CO₂ as 1:3.7 (molar ratio). Porous Pd-HPP-TiO₂ composite with ~65.6 wt% of Pd-HPP shows the CO₂ adsorption capacity of $54 \text{ cm}^3 \text{ g}^{-1}$, corresponding to about 3.2 CO₂ molecules per porphyrin unit. To study the effect of porphyrin concentration on the adsorption of CO₂ and photocatalytic reaction, we have prepared porous Pd-HPP-TiO₂

composites with different mass percentage of porphyrin unit by adjusting the adding amount of porphyrin monomer. The porous Pd-HPP-TiO₂ composites with 53.8 and 74.9 wt% of Pd-HPP display the molar ratio of porphyrin unit/adsorbed CO₂ as 3.1 and 3.0, respectively (Supplementary Fig. 11 and Table S5). The results suggest that the ratio of porphyrin/CO₂ almost keeps constant. A little lower porphyrin/CO₂ ratio in Pd-HPP-TiO₂ composites than that in pure Pd-HPP is presumably due to the introduction of TiO₂ slightly blocking the crosslinking of porphyrin monomer. It can be concluded that the adsorption of CO₂ molecules strongly depends on the porphyrin content. The porphyrin with a core of four pyrrole rings as the building blocks into microporous materials endows them polarizing N-containing groups and large π -conjugated structure, which could response to enhance the interaction with CO₂. The photocatalytic processes for CO₂ reduction involve light absorption to generate electron-hole pairs in TiO₂, electron trapping at the catalytic Pd(II) sites in Pd-HPP, and reduction of the adsorbed CO₂ on Pd(II) sites. That is, both CO₂ adsorption and electron generation and trapping are decisive to the photocatalytic reactions. The result in Supplementary Fig. 12 reveals that there is an appropriate porphyrin content which balanced the CO₂ adsorption and conversion efficiency.

We have now also included the corresponding description on Page 8 in the manuscript. “To study the effect of porphyrin concentration on the adsorption of CO₂ and photocatalytic reaction, we have prepared porous Pd-HPP-TiO₂ composites with different mass percentage of porphyrin unit by adjusting the adding amount of porphyrin monomer. The CO₂ uptake of porous Pd-HPP-TiO₂ composites with 53.8 and 74.9 wt% of Pd-HPP was presented in Supplementary Fig. 11. According to the adding amount of porphyrin monomer and the yield of resulted polymer, the mass percentage of porphyrin unit in Pd-HPP is calculated to be about 70.5%. Thus the molar ratios of porphyrin unit/adsorbed CO₂ can be calculated and compared in Supplementary Table 5. The results suggest that the ratio of porphyrin/CO₂ almost keeps constant. A little lower porphyrin/CO₂ ratio in Pd-HPP-TiO₂ composites than that in pure Pd-HPP is presumably due to the introduction of TiO₂ slightly blocking the crosslinking of porphyrin monomer. It can be concluded that the adsorption of CO₂ molecules strongly

depends on the porphyrin content. Besides the CO₂ adsorption, electron generation on TiO₂ photocatalyst and trapping by Pd(II) sites are crucial processes that involved in the photocatalytic reactions. The result of photocatalytic CO₂ reduction in Supplementary Fig. 12 reveals that there is an appropriate porphyrin content which balanced the CO₂ adsorption and conversion efficiency.”

Table S5. Comparison of the molar ratio of porphyrin/CO₂ in different photocatalysts.

Photocatalysts	Pd-HPP Content (wt%)	Porphyrin Content (wt%)	CO ₂ uptake (cm ³ g ⁻¹) ^[d]	Porphyrin/CO ₂ (molar ratio)
Pd-HPP	100	70.5	96.0	1:3.7
Pd-HPP-TiO ₂ -1 ^[a]	53.8	37.9	42.4	1:3.1
Pd-HPP-TiO ₂ ^[b]	65.6	46.2	54.0	1:3.2
Pd-HPP-TiO ₂ -2 ^[c]	74.9	52.8	58.3	1:3.0

^[a] Pd-HPP-TiO₂-1 was synthesized by adding 15 mg of TPP monomer.

^[b] Pd-HPP-TiO₂ was synthesized by adding 30 mg of TPP monomer.

^[c] Pd-HPP-TiO₂-2 was synthesized by adding 45 mg of TPP monomer.

^[d] Calculated from the CO₂ adsorption-desorption isotherms at 1.00 bar and 273 K.

Supplementary Figure S11. CO₂ adsorption and desorption isotherms of porous Pd-HPP-TiO₂ composites at 273 K. Pd-HPP-TiO₂-1 was synthesized by adding 15 mg of TPP monomer. Pd-HPP-TiO₂-2 was synthesized by adding 45 mg of TPP monomer.

Supplementary Figure S12. The evolution rates of CH₄ and CO over porous Pd-HPP-TiO₂ composites. Pd-HPP-TiO₂-1 was synthesized by adding 15 mg of TPP monomer. Pd-HPP-TiO₂ was synthesized by adding 30 mg of TPP monomer. Pd-HPP-TiO₂-2 was synthesized by adding 45 mg of TPP monomer.

7. At page 9, the authors mentioned that "... the coordination number of Pd in Pd-HPP-TiO₂ is close to 4.0," However, in Pd-K-edge XANES spectrum (Fig. 5(c)), we cannot see any pre-edge peak due to the 4-coordinated species. What is the reason? Is it really 4-coordinated species but not 6-coordinated species?

Response: Thanks for your suggestion. For Pd K-edge XANES spectrum, the pre-edge peak, which originates from the electric-dipole-forbidden 1s→4d transition and can become electric quadrupole allowed through 4d and 5p hybridization, is sensitive to the coordination configuration and symmetry. In general, the planar metal complex exhibits obvious pre-edge peak at the low-energy part of X-ray absorption spectrum (*Sci. Adv.* 2020, 6, eaaz8447; *ACS Appl. Mater. Interfaces* 2019, 11, 44018). In some cases, the pre-edge peak of Pd cannot be observed in Pd-N coordination environment, presumably due to its non-planar configuration (*Adv. Mater.* 2019, 31, 1900509; *Angew. Chem. Int. Ed.* 2021, 60, 345; *Adv. Funct. Mater.* 2020, 30, 2000407). Therefore, the absence of pre-edge peak in Fig. 5c implies the distortion of Pd-N bond deviating from a planar configuration when coordinating with hyper-crosslinking porphyrin-based polymers (HPP). The edge energy of Pd-HPP-TiO₂ is close to that of PdO but higher than that of

Pd foil, confirming Pd(II) in Pd-HPP-TiO₂. The main peak at 1.5 Å in FT-EXAFS spectrum indicates formation of Pd-N bonding (Fig. 5d). The structural parameters based on the well-matched FT-EXAFS fitting reveals the Pd-N₄ coordination environment (Fig. 5e and Supplementary Table 6) (*Adv. Mater.* 2019, 31, 1900509; *Adv. Funct. Mater.* 2020, 30, 2000407).

8. What is the bond distance of Pd-N? It should be obtained from FT-EXAFS spectrum.

Response: Yes, the bond distance of Pd-N can be obtained from FT-EXAFS spectrum. According to the quantitative FT-EXAFS curve fitting, the bond distance (*R*) of Pd-N is 2.03 Å, which has been provided in Supplementary Table 6.

We have now also included the corresponding description on Page 10 in the manuscript. “Supplementary Table 6 reveals that the coordination number of Pd in Pd-HPP-TiO₂ is close to 4.0, indicating that Pd(II) coordinates to four N atoms of the porphyrin. The measured Pd-N bond distance of 2.03 Å is also close to the reported results of Pd-N₄ center^{48,49}.”

9. At page 11, as the authors described, Pd(II) species slowly changes to Pd(0) under UV-visible light irradiation. In fact, as shown in Fig. S15, the formation of CH₄ and CO decrease with increasing irradiation time and also repeating the cycles. Despite these results, the authors mentioned that “indicating the durability of Pd-HPP-TiO₂ for a long-term photocatalytic reaction.” These description is a little rough and rather wrong. This point should be revised correctly, reflecting the real results.

Response: For the reactions using metal nanocatalysts, the durability is a common problem because the metal nanoparticles are inclined to forming aggregations due to their high surface energy, which seriously decreases the catalytic activity. As reported, locking Pd inside the microporous polymers of knitting benzene-triphenylphosphine (Ph-PPh₃) can protect the Pd species from aggregation and precipitation, resulting in the higher catalytic activity and stability for Suzuki–Miyaura coupling reaction than that of PdCl₂ and PdCl₂(PPh₃)₂ catalysts (*Adv. Mater.* 2012, 24, 3390; *Adv. Funct. Mater.* 2020, 31, 2008265). In particular, porphyrin-based polymer has merits of

microporous structure and strong coordination with metal ions, both of which are favorable to stabilizing Pd(II) sites. Therefore, after five cycles, ~79% Pd existed as Pd(II) owing to the coordination to the porphyrin in microporous HPP. Accordingly, Pd-HPP-TiO₂ showed about 80 % of the initial photocatalytic activity for the CO₂ conversion after five consecutive cycles. Although there is somewhat loss in catalytic activity, the microporous porphyrin-based polymer makes great contributions to stabilizing Pd(II) sites. The thermogravimetric and structural analysis further confirmed the considerable stability of Pd-HPP-TiO₂ structure during the photocatalytic reaction.

To make the description more accurate, we have revised the description on Page 12 in the manuscript. “After five cycles, ~79% Pd existed as Pd(II) owing to the coordination to the porphyrin in microporous HPP, indicating the considerable stability of Pd-HPP-TiO₂ structure during the photocatalytic reaction.”

10. Regarding the reaction mechanism in Fig. 1, the authors should be discussed the details on the following points. For the formation of CH₄, it is necessary to be formed the bare C from CO₂ via CO. The mechanism for the formation of bare C should be discussed. And also, for the formation of CH₄, H atoms are required. This point should also be discussed.

Response: We appreciate the reviewer’s kind suggestion. According to the detailed studies on the mechanism of CO₂ reduction, there are two possible pathways, i.e. one is the carbene pathway (CO₂ → CO → •C → •CH₃ → CH₄) and the other is the formaldehyde pathway (CO₂ → HCOOH → CH₂O → CH₃OH → CH₄) (*Chem. Sci.* 2021, 12, 4267; *Angew Chem Int Ed.* 2013, 52, 7372; *ACS Catal.* 2016, 6, 2018). Although formaldehyde and methanol have been reported as products in some setups, they are not detected in this work. We used acetic ether to wash the photocatalyst after photoreduction and extract less-volatile products. The extracted solution was tested by GC-MS and there was no signal of other products. This result is reasonable because the procedure conducted in gas-solid system can normally form CO and CH₄ rather than formaldehyde and methanol products. Therefore, the CO₂ reduction pathway is more likely to be carbene pathway in the gas-solid system.

We have now also included the corresponding description on Pages 11-12 in the manuscript. “According to the detailed studies on the mechanism of CO₂ reduction, there are two possible pathways, i.e. one is the formaldehyde pathway and the other is the carbene pathway⁵⁸⁻⁶⁰. Although formaldehyde and methanol have been reported as products in some setups, they are not detected in this work. The photocatalytic CO₂ reduction in the gas-solid reaction can normally form CO and CH₄.^{25,26} Therefore, the CO₂ reduction is more likely to be a carbene pathway as CO₂ → CO → •C → •CH₃ → CH₄. The CO₂ molecules are activated at Pd(II) sites in Pd-HPP and then react with the proton and electron to produce the intermediate *CO. At this time, part of CO is produced by *CO desorption. Since Pd can also function efficiently for the decomposition of H₂O to generate Pd-H species^{61,62}, the adsorbed *CO on Pd will be further react with the dissociated H to the formation of •C radicals, which can successively combine with •H radicals, thereby forming •CH, •CH₂, •CH₃, and finally CH₄ product^{53,63-65}.”

58. Rawool, S. A. et al. Defective TiO₂ for photocatalytic CO₂ conversion to fuels and chemicals. *Chem Sci.* **12**, 4267 (2021).

59. Habisreutinger, S. N. et al. Photocatalytic reduction of CO₂ on TiO₂ and other semiconductors. *Angew Chem. Int. Ed.* **52**, 7372-7408 (2013).

60. Ji, Y. F. et al. Theoretical study on the mechanism of photoreduction of CO₂ to CH₄ on the anatase TiO₂(101) surface. *ACS Catal.* **6**, 2018-2025 (2016).

61. Gao, D. et al. Pd-containing nanostructures for electrochemical CO₂ reduction reaction. *ACS Catal.* **8**, 1510-1519 (2018).

62. Fan, Z. et al. Synthesis of 4H/fcc noble multimetallic nanoribbons for electrocatalytic hydrogen evolution reaction. *J. Am. Chem. Soc.* **138**, 1414 (2016).

63. Anpo, M. et al. Photocatalytic reduction of CO₂ with H₂O on various titanium oxide catalysts. *J. Electroanal. Chem.* **396**, 21-26 (1995).

64. Fan, J. et al. Insight into synergetic mechanism of Au@Pd and oxygen vacancy sites for coupling light-driven H₂O oxidation and CO₂ reduction. *J. Catal.*, **378**, 164-175 (2019).

65. Liu, J. et al. Mechanism of CO₂ photocatalytic reduction to methane and methanol

on defected anatase TiO₂ (101): A DFT study. *J. Phys. Chem. C*, **123**, 3505-3511 (2019).

Reviewer #2:

The manuscript by Ma et al. describes the reduction of CO₂ in air to CO and CH₄ over a Pd-HPP-TiO₂ catalyst. The direct conversion of CO₂ in flue gas or air to fuels is vital. However, the reduction of CO₂ to C1 products have already been extensively studied and the C1 yield of this work is low. Besides, the catalyst material is too complicated. The functions of different parts of the composite: Pd, HPP and TiO₂, and their synergistic interactions are not clearly explained. Therefore, the manuscript is ill-suited to be published in Nature Communications. Other comments are listed below:

Response: Thanks much for appreciating the idea of work presented in this manuscript. Direct photocatalytic CO₂ reduction from flue gas or air is highly desired in practice, but seriously impeded by O₂ reduction as the thermodynamically favorable process. To the best of our knowledge, this is the first example that realized photocatalytic CO₂ reduction in an aerobic environment by taking the advantage of selective CO₂ adsorption in microporous polymer. Owing to the improved CO₂ adsorption capability and selectivity by HPP, the photocatalytic CO₂ conversion over Pd-HPP-TiO₂ can reach as high as 12 % of CO₂ from air after 2 h UV-visible light irradiation just using H₂O as electron donor, which is the highest value among the reported results from air (Supplementary Table 1).

In addition, the evolution rates of C1 products over Pd-HPP-TiO₂ from photocatalytic reduction in pure CO₂ are 48.0 and 34.0 μmol g⁻¹ h⁻¹ for CH₄ and CO, respectively. The photocatalytic reaction of CO₂ with H₂O takes place under mild gas–solid reaction conditions without the use of Ru-containing photosensitizer or sacrificial reagent. Although there are some reports achieving relatively high yields of C1 products by photocatalytic CO₂ reduction, they generally require the addition of Ru-containing photosensitizer together with organic sacrificial reagent and solvent, which present unsustainable and negative environmental impact issues. The reported results under similar reaction conditions (gas-solid reaction just using H₂O as electron donor) have

been provided in Supplementary Table 1, suggesting the excellent photocatalytic activity of the designed photocatalyst in this work.

For functions of Pd, HPP and TiO₂ in the Pd-HPP-TiO₂ composite, here we would like to make an explanation as follows. The composite structure is presented as building Pd(II) sites into CO₂-adsorptive HPP on hollow TiO₂ surface. The microporous HPP can not only improve the capability and selectivity of CO₂ adsorption in an aerobic environment but also stabilize Pd(II) sites. Thus, CO₂ molecules can be selectively enriched in Pd-HPP. Under light irradiation, the photogenerated electrons in the conduction band of TiO₂ transfer efficiently to Pd-HPP and are trapped at the coordinated Pd(II) sites in Pd-HPP to react with the adsorbed CO₂ to produce CH₄ and CO. The holes in the valence band of TiO₂ can oxidize water to produce O₂. The schematic illustration of photocatalytic CO₂ reduction over Pd-HPP-TiO₂ composite is displayed in Figure 1. We are hoping that the functions of Pd, HPP and TiO₂ in the Pd-HPP-TiO₂ composite could be clearly understandable.

We also thank the reviewer's kind suggestions. We have addressed all the issues point-by-point in our response below.

1. The evolution rates for CO and CH₄ are relatively low and the reduction of CO₂ is not selective to a single product.

Response: While we appreciate the reviewer's comments, we want to clarify that selective photocatalytic CO₂ reduction in aerobic environment is highly desired in practice due to its possibility of controlling CO₂ emission from flue gas and reduce CO₂ concentration in air. However in air or flue gas, the CO₂ adsorption and activation on the surface of photocatalysts are low, due to the competitive O₂ adsorption and reduction, as well as the low CO₂ concentration (*J. Am. Chem. Soc.* 2019, 141, 5267; *Sci. Bull.* 2019, 64, 1890; *Green Chem.* 2017, 19, 5777). To the best of our knowledge, this is the first example that realized photocatalytic CO₂ reduction in an aerobic environment by taking the advantage of selective CO₂ adsorption in microporous polymer. Owing to the improved CO₂ adsorption capability and selectivity by HPP, the photocatalytic CO₂ conversion over Pd-HPP-TiO₂ can reach as high as 12 % of CO₂

from air after 2 h UV-visible light irradiation just using H₂O as electron donor, which is the highest value among the reported results from air (Supplementary Table 2).

In addition, the evolution rates of CH₄ and CO over Pd-HPP-TiO₂ in pure CO₂ are 48.0 and 34.0 μmol g⁻¹ h⁻¹, respectively. The photocatalytic reaction of CO₂ with H₂O takes place under mild gas–solid reaction conditions without the use of Ru-containing photosensitizer or sacrificial reagent. Although there are some reports achieving relatively high product yields by photocatalytic CO₂ reduction, they generally require the addition of Ru-containing photosensitizer together with organic sacrificial reagent and solvent, which present unsustainable and negative environmental impact issues. The reported results under similar reaction conditions (gas-solid reaction just using H₂O as electron donor) have been provided in Supplementary Table 1, suggesting the excellent photocatalytic activity of the designed photocatalyst in this work.

The main products during the photocatalytic reaction are CH₄ and CO from CO₂ reduction and O₂ from H₂O oxidation. Normally, CH₄ has much higher calorific value than CO but its evolution is relatively difficult since the adsorbed CO₂ molecules should accept eight electrons and eight protons to break the C–O bonds and form the C–H bonds (*ACS Catal.* 2016, 6, 2018). In contrast, the adsorbed CO₂ molecules are more readily converted to CO than CH₄ on the surface of semiconductor photocatalysts due to the release of *CO intermediate from photocatalyst surface before it can accept the subsequent electrons to be further reduced (*J. Am. Chem. Soc.* 2017, 139, 5660; *J. Am. Chem. Soc.* 2021, 143, 2984). In this work, the higher evolution rate of CH₄ than CO can be attributed to the enriched CO₂ molecules in Pd-HPP and efficient electron trapping at Pd(II) sites. As suggested, the selective reduction of CO₂ to CH₄ and more valuable products is worthy of deep study in our future work.

We have now included the corresponding description and the literature on Page 4 in the manuscript. “When building Pd(II) sites into HPP-TiO₂, the CO₂ reduction efficiency was further enhanced, reaching high evolution rates of 48.0 and 34.0 μmol g⁻¹ h⁻¹ (average value within 4 h) for CH₄ and CO, respectively. The comparison to the reported results under similar reaction conditions suggests the excellent photocatalytic activity of porous Pd-HPP-TiO₂ composite (Supplementary Table 1)”

Table S1. Comparison of the photocatalytic activity of Pd-HPP-TiO₂ with the recently reported catalysts for CO₂ reduction under similar reaction conditions (gas-solid reaction just using H₂O as electron donor).

Photocatalysts	Products ($\mu\text{mol g}^{-1} \text{h}^{-1}$)		Ref
	CH ₄	CO	
Pyrazolyl Porphyrinic Ni-MOF	10.1	6.0	5
1%B/g-C ₃ N ₄	0.16	-	6
α -Fe ₂ O ₃ /Amine-RGO/CsPbBr ₃	28.5	6.0	7
V _o -BiOIO ₃	-	16.33	8
N-Doped Graphene on CdS	0.33	2.59	9
V _s -CuIn ₅ S ₈	8.7	-	10
Porous CoO@N-GCs	10.03	5.16	11
Hypercrosslinked Polymers-3	0.30	5.10	12
Porphyrin Based COF	-	24.6	13
Triphenylamine Based CMPs	-	37.15	14
ZnSe/CdS DORs	-	11.3	15
Pd-HPP-TiO ₂	48	34	This work

- Fang, Z. *et al.* Boosting interfacial charge-transfer kinetics for efficient overall CO₂ photoreduction via rational design of coordination spheres on metal–organic frameworks *J. Am. Chem. Soc.* **142**, 12515-12523 (2020).
- Fu, J. *et al.* Graphitic carbon nitride with dopant induced charge localization for enhanced photoreduction of CO₂ to CH₄. *Adv. Sci.* **6**, 1900796 (2019).
- Jiang, Y. *et al.* All-solid-state Z-scheme α -Fe₂O₃/amine-RGO/CsPbBr₃ hybrids for visible light-driven photocatalytic CO₂ reduction. *Chem* **6**, 766-780 (2020).
- Chen, F. *et al.* Macroscopic spontaneous polarization and surface oxygen vacancies collaboratively boosting CO₂ photoreduction on BiOIO₃ single crystals. *Adv. Mater.* **32**, 1908350 (2020).
- Bie, C. *et al.* In situ grown monolayer N-doped graphene on CdS hollow spheres with seamless contact for photocatalytic CO₂ reduction. *Adv. Mater.* **31**, 1902868 (2019).
- Li, X. *et al.* Selective visible-light-driven photocatalytic CO₂ reduction to CH₄ mediated by atomically thin CuIn₅S₈ layers. *Nat. Energy* **4**, 690-699 (2019).
- He, L., Zhang, W., Liu, S. & Zhao, Y. Three-dimensional porous N-doped graphitic

carbon framework with embedded CoO for photocatalytic CO₂ reduction. *Appl. Catal. B: Environ.* **298**, 120546 (2021).

12. Schukraft, G. E. M. *et al.* Hypercrosslinked polymers as a photocatalytic platform for visible-light-driven CO₂ photoreduction using H₂O. *ChemSusChem* **14**, 1720-1727 (2021).

13. Wang, L. *et al.* Improved photoreduction of CO₂ with water by tuning the valence band of covalent organic frameworks. *ChemSusChem* **13**, 2973-2980 (2020).

14. Dai, C. *et al.* Triphenylamine based conjugated microporous polymers for selective photoreduction of CO₂ to CO under visible light. *Green Chem.* **21**, 6606-6610 (2019).

15. Xin, Z. *et al.* Rational design of dot-on-rod nanoheterostructure for photocatalytic CO₂ reduction: pivotal role of hole transfer and utilization. *Adv. Mater.* DOI: <https://doi.org/10.1002/adma.202106662>.

2. Some language errors should be corrected in this manuscript, such as “followed by loading Pd(II) to via coordination with HPP” in the introduction section.

Response: We are sorry for the mistakes. We have carefully checked the whole manuscript and corrected the language errors. The corrections in the manuscript are marked in red colour.

3. In the introduction section, “Pd allows us to confirm the influence on the reduction.” What influence? the authors should clarify the meanings of the sentence.

Response: We choose Pd as CO₂ reduction site due to its enhancement in photocatalytic activity and CH₄ selectivity from CO₂ reduction (*Chem. Eng. J.* 2021, **410**, 128234; *J. Am. Chem. Soc.* 2017, 139, 4486). Since the composite structure is presented as building catalytic sites into CO₂-adsorptive HPP on hollow TiO₂ surface, the choice of Pd allows us to confirm the influence of CO₂ adsorption and charge separation on the reduction.

According to the reviewer’s suggestion, we have revised the description on Page 3 in the manuscript. “The choice of Pd allows us to confirm the influence of CO₂ adsorption and charge separation on the reduction.”

4. As shown in Supplementary Fig. S5, after the addition of HPP, the composite showed favourable absorption in the visible light region. Why are not the CO₂ photoreduction experiments carried out under visible light irradiation?

Response: This is a good suggestion. Porphyrin-based polymers displayed strong absorption in a wide region of visible light (Supplementary Fig. 6), so we have conducted the photocatalytic CO₂ reduction under visible light irradiation. As shown in Supplementary Fig. 7, both Pd-HPP and Pd-HPP-TiO₂ exhibit low photocatalytic activity for CO₂ reduction under visible light irradiation, suggesting the low efficiency of charge separation in HPP (*ChemSusChem* 2021, 14, 1720). The comparison of visible-light driven CO₂ reduction to that under UV-visible light has been presented in Supplementary Fig. 8. It can be deduced that the photocatalytic CO₂ reduction reaction over Pd-HPP-TiO₂ dominantly depended on UV light of UV-visible light irradiation. Generally, the photoinduced charge separation in organic polymers does not occur dominantly compared with exciton migration, leading to the lower capability as redox photocatalysts (*Angew. Chem. Int. Ed.* 2019, 58, 10236). The extremely low photocurrent of Pd-HPP in Fig. 2c further revealed its deficiency in providing available electrons under light irradiation. The visible light is absorbed by HPP, and most of photons absorbed are changed to heat.

The corresponding description has been provided on Page 6 in the manuscript as: “Although Pd-HPP possesses strong absorption in the visible region (Supplementary Fig. 5), both Pd-HPP and Pd-HPP-TiO₂ exhibit low photocatalytic activity for CO₂ reduction under visible light irradiation (Supplementary Fig. 7), suggesting the low efficiency of charge separation in HPP. It is found that such results are comparable to the recently reported analogous polymer photocatalyst³⁸. The comparison of visible-light driven CO₂ reduction to that under UV-visible light is presented in Supplementary Fig. 8. Thus, the photocatalytic CO₂ reduction reaction over Pd-HPP-TiO₂ depended on UV light of UV-visible light irradiation. The visible light is absorbed by HPP, and most of photons absorbed are changed to heat.”

38. Schukraft, G. E. M. et al. Hypercrosslinked polymers as a photocatalytic platform for visible light-driven CO₂ photoreduction using H₂O. *ChemSusChem* **14**, 1720-1727

(2021).

Supplementary Figure S8. The evolution rates of CH₄ and CO from CO₂ reduction over Pd-HPP-TiO₂ photocatalyst under visible light irradiation ($\lambda \geq 420$ nm) and full light irradiation. (a) in pure CO₂ (b) in CO₂ with 2 vol% O₂.

5. The CO₂ photoreduction testing time is too short (only 4 h). And the XRD pattern of Pd-HPP-TiO₂ after the cycling test was changed, indicating that the stability of the catalyst is unsatisfactory.

Response: The photocatalytic reaction of CO₂ reduction normally takes place in a batch reactor, so the reduction products will accumulate in the reactor and partly adsorbed on the photocatalyst surface, leading to the decreased catalytic activity. In general, the CO₂ photoreduction testing is conducted in a time period of several hours (*Chem. Mater.* 2020, 32, 1517; *Angew. Chem. Int. Ed.* 2020, 59, 2.). According to the reviewer's suggestion, we have tested the CO₂ photoreduction experiment for longer time (up to 20 h) to evaluate the stability of the catalyst. Pd-HPP-TiO₂ showed continuous CH₄ and CO production up to 20 h under UV-visible light irradiation (Supplementary Fig. 1). Although there is somewhat loss in catalytic activity, the superior performance of porous Pd-HPP-TiO₂ to Pd/TiO₂ during long-term photocatalytic reaction suggests that the introduction of microporous HPP greatly contributes to stabilizing Pd(II) sites. In addition, the structural analysis and cycling test further confirmed the considerable stability of Pd-HPP-TiO₂ structure during the photocatalytic reaction.

Actually, for the reactions using metal nanocatalysts, the durability is a common problem because the metal nanoparticles are inclined to forming aggregations due to their high surface energy, which seriously decreases the catalytic activity. As reported,

locking Pd inside the microporous polymers of knitting benzene-triphenylphosphine (Ph-PPh₃) can protect the Pd species from aggregation and precipitation, resulting in the higher catalytic activity and stability for Suzuki–Miyaura coupling reaction than that of PdCl₂ and PdCl₂(PPh₃)₂ catalysts (*Adv. Mater.* 2012, 24, 3390; *Adv. Funct. Mater.* 2020, 31, 2008265.). In particular, porphyrin-based polymer has merits of microporous structure and strong coordination with metal ions, both of which are favorable to stabilizing Pd(II) sites.

The corresponding description has been included on Page 4 in the manuscript as: “In a long-term test, Pd-HPP-TiO₂ showed continuous CH₄ and CO production up to 20 h under UV-visible light irradiation (Supplementary Fig. 1). Although there is somewhat loss in catalytic activity, the superior performance of porous Pd-HPP-TiO₂ to Pd/TiO₂ during long-term photocatalytic reaction suggests that the introduction of microporous HPP greatly contributes to stabilizing Pd(II) sites.”

27. Blommaerts, N. et al. Tuning the turnover frequency and selectivity of photocatalytic CO₂ reduction to CO and methane using platinum and palladium nanoparticles on Ti-Beta zeolites. *Chem. Eng. J* **410**, 128234 (2021).

Supplementary Figure S1. Long-term photocatalytic CO₂ reduction over Pd-HPP-TiO₂ (a) and Pd/TiO₂ (b) photocatalysts.

Reviewer #3:

The authors report on the preparation, characterization and testing of a Pd-HPP-TiO₂

photocatalyst, that enables effective CO₂ to methane conversion, even in the presence of oxygen gas, due to a preferential adsorption of CO₂ over O₂ in the material. The concept of developing O₂-tolerant CO₂ conversion photocatalysts is noteworthy, considering applications in e.g. flue gas treatment, and is thus expected to attract attention from a broad audience. The work in itself is quite solid and very complete. All proper benchmark experiments have been carried out (e.g. dark reference, Pd directly on TiO₂, Pd-HPP alone, etc.). The preferential and quantitative adsorption of CO₂ on the Pd-HPP-TiO₂ catalyst is demonstrated well, as well as its chemical structure (i.e. indeed Pd(II) instead of Pd(0) NPs, coupled to the polymer). Also, the improved charge transfer dynamics have been characterized appropriately using electrochemical tools. Hence, I recommend publication of this manuscript

Response: First, we would like to thank the reviewer to realize and appreciate the importance of work reported in this manuscript. We also thank the reviewer's overall comments and kind suggestions that helped us to improve the manuscript significantly. We have now addressed the reviewer's comments point-wise in our response below.

1. at the bottom of page 2 the authors mention an anaerobic environment, while I believe it should be aerobic.

Response: Thanks for the reviewer's kind suggestion. We have revised this sentence on Page 2 in the manuscript as: "For the photocatalytic CO₂ reduction with H₂O in such an aerobic environment, another essential requirement is to assemble the photocatalytic sites for CO₂ reduction and H₂O oxidation for efficient separation of photogenerated electrons and holes, respectively."

2. Only a Xe lamp is mentioned in the method section as the light source, while the text also mentions AM 1.5 simulated solar light. Have any filters been used? How is the absolute irradiance at sample distance controlled and experimentally verified?

Response: In this work, only a 300 W Xe lamp is used as the light source. No filter is used to ensure the photocatalytic reaction under UV-visible light irradiation. The full spectrum of the Xe lamp is close to that of solar light in the UV-visible light region

(325~780 nm), as shown in Supplementary Fig. 24. For the visible-light driven experiment, a cut-off filter of 420 nm was equipped with the lamp. The powder photocatalyst was dispersed on the middle of the culture ware, placed in the sealed custom-made glass vessel, 10 cm away from the light source. The irradiance is quantified using the "average light intensity", which is calculated by averaging the measured values of four corners of samples.

On Page 15, we have added the following content in the method section: “The reaction mixture was irradiated using a 300 W Xenon lamp (PLS-SXE300D, Beijing Perfectlight Technology, China) as light source. The full spectrum locates in the UV-visible light region (325~780 nm), as shown in Supplementary Fig. 24. For the visible-light driven experiment, a cut-off filter of 420 nm was equipped with the lamp.”

Supplementary Figure S24. The full spectrum of the Xe lamp (PLS-SXE300D, Beijing Perfectlight, China).

3. The CO₂ conversion selectivity of photocatalysts containing Pd-species towards methane could be further corroborated with recent literature (e.g. Chemical Engineering Journal 410 (2021) 128234)

Response: We appreciate the reviewer’s kind suggestion. In this work, Pd was chose as CO₂ reduction site due to its enhancement in photocatalytic activity and CH₄ selectivity from CO₂ reduction.

We have now included the corresponding description and the literature on Page 4 in

the manuscript. “When building Pd(II) sites into HPP-TiO₂, the CO₂ reduction efficiency was further enhanced, reaching high evolution rates of 48.0 and 34.0 μmol g⁻¹ h⁻¹ for CH₄ and CO, respectively...The high selectivity as 59% for CH₄ production over Pd-HPP-TiO₂ photocatalyst can be attributed to the Pd(II) sites with sufficient energy overcoming the Schottky barrier with TiO₂ and thus improving the charge separation efficiency²⁷.”

27. Blommaerts, N. et al. Tuning the turnover frequency and selectivity of photocatalytic CO₂ reduction to CO and methane using platinum and palladium nanoparticles on Ti-Beta zeolites. *Chem. Eng. J* **410**, 128234 (2021).

4. On p. 6 it is hypothesized that visible light absorption is mainly converted into heat, while UV absorption by TiO₂ truly results in the observed activity after charge transfer. While I, too, believe this is most probable, has it actually been verified experimentally? For instance, a UV cut-off filter at 420 nm could be used to verify the absence/presence of purely visible light photoactivity?

Response: Yes, a UV cut-off filter at 420 nm could be used to verify the presence of purely visible light photoactivity. Since porphyrin-based polymers displayed strong absorption in a wide region of visible light (Supplementary Fig. 6), we have tested the visible-light driven photocatalytic CO₂ reduction using a Xe lamp equipped with a cut-off filter of 420 nm. As shown in Supplementary Fig. 7, both Pd-HPP and Pd-HPP-TiO₂ exhibit low photocatalytic activity for CO₂ reduction under visible light irradiation, suggesting the low efficiency of charge separation in HPP (*ChemSusChem* 2021, 14, 1720). The comparison of visible-light driven CO₂ reduction to that under UV-visible light has been presented in Supplementary Fig. 8. It can be deduced that the photocatalytic CO₂ reduction reaction over Pd-HPP-TiO₂ dominantly depended on UV light of UV-visible light irradiation. The extremely low photocurrent of Pd-HPP in Fig. 2c further revealed its deficiency in providing available electrons under light irradiation. The visible light is absorbed by HPP, and most of photons absorbed are changed to heat.

The corresponding description has been provided on Pages 6 and 15 in the manuscript as: “Although Pd-HPP possesses strong absorption in the visible region

(Supplementary Fig. 5), both Pd-HPP and Pd-HPP-TiO₂ exhibit low photocatalytic activity for CO₂ reduction under visible light irradiation (Supplementary Fig. 6), suggesting the low efficiency of charge separation in HPP. It is found that such results are comparable to the recently reported analogous polymer photocatalyst³⁸. The comparison of visible-light driven CO₂ reduction to that under UV-visible light is presented in Supplementary Fig. 8. Thus, the photocatalytic CO₂ reduction reaction over Pd-HPP-TiO₂ depended on UV light of UV-visible light irradiation. The visible light is absorbed by HPP, and most of photons absorbed are changed to heat.”

“For the visible-light driven experiment, a cut-off filter of 420 nm was equipped with the lamp.”

38. Schukraft, G. E. M. et al. Hypercrosslinked polymers as a photocatalytic platform for visible light-driven CO₂ photoreduction using H₂O. *ChemSusChem* **14**, 1720-1727 (2021).

Supplementary Figure S8. The evolution rates of CH₄ and CO from CO₂ reduction over Pd-HPP-TiO₂ photocatalyst under visible light irradiation ($\lambda \geq 420$ nm) and full light irradiation. (a) in pure CO₂ (b) in CO₂ with 2 vol% O₂.

5. On page 7 a type I isotherm is mentioned, which does not correspond with the fact that a H₄ type hysteresis loop is observed (characteristic for hollow spheres with walls composed of ordered mesoporous material). It is therefore rather a type IV isotherm.

Response: We appreciate the reviewer’s kind suggestion. Pure hollow TiO₂ possesses a typical mesoporous structure, which is confirmed by a type IV isotherm with a H₄ hysteresis loop at medium pressure. The N₂ adsorption-desorption isotherms of Pd-HPP and Pd-HPP-TiO₂ have the mixed character of type I and IV, exhibiting a steep increase

at relative low pressure ($P/P_0 < 0.001$) and an obvious hysteresis at medium pressure, which indicate the existence of abundant micropores and mesopores.

We have revised them on Page 8 in the revised manuscript. “As shown in Fig. 3a, the N_2 adsorption-desorption isotherms of Pd-HPP and Pd-HPP- TiO_2 exhibit a steep increase at relative low pressure ($P/P_0 < 0.001$) and an obvious hysteresis at medium pressure, which indicate the existence of abundant micropores and mesopores⁴². This result may be due to a fast rate of hyper-crosslinking and the low degree of free packing for building blocks by Friedel-Crafts alkylation reaction. In contrast, pure hollow TiO_2 shows the character of type IV isotherm with a hysteresis loop at medium pressure, which suggests the formation of mesoporous structure and gives a Brunauer–Emmett–Teller surface area (S_{BET}) of $75 \text{ m}^2 \text{ g}^{-1}$.”

6. It would be good to improve the overall level of English, as the text contains quite some (grammatical) errors, which tend to distract the reader's attention from the actual nice experimental work.

Response: Following the reviewers’ suggestions, we have carefully checked the whole manuscript and refined the language including the corrections of several grammatical errors. The corrections in the manuscript are marked in red colour.

REVIEWER COMMENTS

Reviewer #1 (Remarks to the Author):

This revised version is well written and polished up in accordance with the suggestions by Referees. This revised MS is now acceptable for publication in Nat. Commun. in the present state.

Reviewer #2 (Remarks to the Author):

While I appreciate the efforts that the authors put into reviewing the manuscript and answering my comments, I still believe critical statements and discussions about the in situ DRIFTS spectra and the photocatalytic performances should be included in the paper.

1. For the in situ DRIFTS spectra in Fig. 5f, if the peak at 1740 cm^{-1} can be ascribed to surface adsorbed carbonate species, why did the peak change to negative under light irradiation? Even if the carbonate species are completely consumed, the spectra only become flat. The authors should explain this phenomenon more carefully.
2. In Fig.2, only one measurement is not very convincing, the authors should measure the photocatalytic performances of the catalysts for multiple times and provide average values with error bars.
3. The authors discussed that the reaction pathway was $\text{CO}_2 \rightarrow \text{CO} \rightarrow \bullet\text{C} \rightarrow \bullet\text{CH}_3 \rightarrow \text{CH}_4$. However, COOH^* is a vital intermediate during CO_2 conversion to CO^* . Why didn't the authors detect COOH^* in the in situ DRIFTS spectra?

Given below are our responses (in **BLUE** colour) to the reviewer' comments. The changes to the manuscript and supplementary information are marked in **RED** colour.

Reviewer #1

This revised version is well written and polished up in accordance with the suggestions by Referees. This revised MS is now acceptable for publication in Nat. Commun. in the present state.

Response: We thank the reviewer for appreciating the improvement achieved in this revised version. We also appreciate the reviewers' recommendation of our manuscript for publication in Nature Communications.

Reviewer #2:

While I appreciate the efforts that the authors put into reviewing the manuscript and answering my comments, I still believe critical statements and discussions about the in situ DRIFTS spectra and the photocatalytic performances should be included in the paper.

Response: We thank the reviewer for appreciating the efforts that we made in this revised version. We also appreciate the reviewers' comments and suggestions that helped us to improve the manuscript. We have addressed the reviewers' comments point-wise in our response below:

1. For the in situ DRIFTS spectra in Fig. 5f, if the peak at 1740 cm⁻¹ can be ascribed to surface adsorbed carbonate species, why did the peak change to negative under light irradiation? Even if the carbonate species are completely consumed, the spectra only become flat. The authors should explain this phenomenon more carefully.

Response: Thanks for your suggestion. As shown in Fig. 5f, the infrared spectrum of Pd-HPP-TiO₂ photocatalyst at 0 min in the dark was collected in Ar environment, which was set as the baseline. Actually, the surface adsorbed carbonate species cannot be removed completely by blowing Ar gas due to the high CO₂ adsorption of Pd-HPP-TiO₂ (54.0 cm³ g⁻¹, Fig. 3c). That is, the infrared absorption of the surface adsorbed carbonate species still exists even though they cannot be shown in the baseline. Under light irradiation, the peaks of the surface adsorbed carbonate species were significantly weakened. Especially, the peak at 1740 cm⁻¹ first became flat at 2 min and then changed to a negative peak with prolonged irradiation. The results indicated that the adsorbed carbonate species on Pd-HPP-TiO₂ surface could be efficiently consumed during the photocatalytic reaction. Similar phenomena of in situ DRIFTS experiments during CO₂ reduction can also be observed in many literatures (*Adv. Mater.* 2021, 2105135; *Appl. Catal. B-Environ.* 2021, 284, 119692; *ACS Nano* 2020, 14, 13103; *Adv. Funct. Mater.* 2019, 29, 1905153; *Adv. Mater.* 2019, 31, 1902868).

According to the suggestion, we have added the corresponding description on Page 11 in the manuscript. “Under UV-visible light irradiation, the peaks at 1690 and 1640 cm⁻¹ were significantly weakened. Meanwhile, the peak at 1740 cm⁻¹ first became flat at 2 min and then changed to a negative peak with prolonged irradiation, which could be explained by the existence of pre-adsorbed carbonate species on the Pd-HPP-TiO₂ surface before collecting the baseline due to its high CO₂ uptake.”

2. In Fig.2, only one measurement is not very convincing, the authors should measure the photocatalytic performances of the catalysts for multiple times and provide average values with error bars.

Response: As suggested, we have measured the photocatalytic performances of the catalysts for three times. Fig. 2(a, d, e, f) were updated in the revised manuscript to provide the average values with error bars, which confirmed the repeatable results.

Fig. 2 Evaluation of photocatalytic CO₂ reduction. (a) The evolution rates of CH₄ and CO in pure CO₂. (b) On-line monitoring of O₂ evolution during the photocatalytic reaction over Pd-HPP-TiO₂. (c) Photocurrent response during light on-off cycling. (d) Comparison of the CH₄ evolution rates in pure CO₂ and air. (e) The conversion yield of CO₂ by measuring the CO₂ concentration. (f) Effect of O₂ concentration (vol%) in CO₂/O₂ gas mixture on the CH₄ evolution rate. The results in (a, d, e, f) are the average values of three parallel experiments. The error bar represents the standard deviation of the measurements.

3. The authors discussed that the reaction pathway was $\text{CO}_2 \rightarrow \text{CO} \rightarrow \cdot\text{C} \rightarrow \cdot\text{CH}_3 \rightarrow \text{CH}_4$. However, COOH^* is a vital intermediate during CO₂ conversion to CO*. Why didn't the authors detect COOH^* in the in situ DRIFTS spectra?

Response: Thanks for your suggestion. COOH^* is a vital intermediate during CO₂ conversion to CO*. According to many recent literatures, the new peak at 1589 cm⁻¹ emerged in the spectra during light irradiation should be ascribed to the C=O stretching vibration of *COOH groups, which was the vital intermediate for *CO formation and then transformed to CO and other fuels (*J. Am. Chem. Soc.* 2021, 143, 18233; *Adv. Mater.* 2021, 33, 2100143; *Adv. Energy Mater.* 2021, 11, 2102389; *ACS Nano* 2021, 15, 14453; *Nat. Commun.* 2020, 11, 1149).

According to the suggestion, we have revised the corresponding description on Page 11 in the manuscript. “Meanwhile, a new peak at 1589 cm^{-1} emerged in the spectra is ascribed to the C=O stretching vibration of *COOH groups, which was the vital intermediate for *CO formation and then transformed to CO and other fuels.⁵⁶⁻⁵⁸...Therefore, the CO₂ reduction is more likely to be a carbene pathway as CO₂ → COOH → CO → •C → •CH₃ → CH₄. The CO₂ molecules are activated at Pd(II) sites in Pd-HPP and then react with the proton and electron to produce the intermediate *COOH. Subsequent reaction with the electron and proton results in the splitting of *COOH into *CO and H₂O.”

56. Zhu, J. et al. Asymmetric triple-atom sites confined in ternary oxide enabling selective CO₂ photothermal reduction to acetate. *J. Am. Chem. Soc.* **143**, 18233–18241 (2021).

57. Di, J. et al. Surface local polarization induced by bismuth-oxygen vacancy pairs tuning non-covalent interaction for CO₂ photoreduction. *Adv. Energy Mater.* **11**, 2102389 (2021).

58. Wang, S. et al. Intermolecular cascaded π -conjugation channels for electron delivery powering CO₂ photoreduction. *Nat. Commun.* **11**, 1149 (2020).